# Modification of *BRCA1*-associated breast cancer risk by HMMR overexpression

Francesca Mateo[1,15], Zhengcheng He[2,15], Lin Mei[2,15], Gorka Ruiz de Garibay[1], Carmen Herranz[1], Nadia García[1], Amanda Lorentzian[2], Alexandra Baiges[1], Eline Blommaert[1], Antonio Gómez[3], Oriol Mirallas[1], Anna Garrido-Utrilla[1], Luis Palomero[1], Roderic Espín[1], Ana I. Extremera[1], M. Teresa Soler-Monsó[4], Anna Petit[4], Rong Li[5], Joan Brunet[6], Ke Chen[2], Susanna Tan[7], Connie J. Eaves[7], Curtis McCloskey[8], Razq Hakem[8,9], Rama Khokha[8,9], Philipp F. Lange[10,11], Conxi Lázaro[12,13], Christopher A. Maxwell[2,11✉] & Miquel Angel Pujana[1,14✉]

Breast cancer risk for carriers of *BRCA1* pathological variants is modified by genetic factors. Genetic variation in *HMMR* may contribute to this effect. However, the impact of risk modifiers on cancer biology remains undetermined and the biological basis of increased risk is poorly understood. Here, we depict an interplay of molecular, cellular, and tissue microenvironment alterations that increase *BRCA1*-associated breast cancer risk. Analysis of genome-wide association results suggests that diverse biological processes, including links to *BRCA1-HMMR* profiles, influence risk. HMMR overexpression in mouse mammary epithelium increases *Brca1*-mutant tumorigenesis by modulating the cancer cell phenotype and tumor microenvironment. Elevated HMMR activates AURKA and reduces ARPC2 localization in the mitotic cell cortex, which is correlated with micronucleation and activation of cGAS-STING and non-canonical NF-κB signaling. The initial tumorigenic events are genomic instability, epithelial-to-mesenchymal transition, and tissue infiltration of tumor-associated macrophages. The findings reveal a biological foundation for increased risk of *BRCA1*-associated breast cancer.

[1] ProCURE, Catalan Institute of Oncology, Oncobell, Bellvitge Institute for Biomedical Research (IDIBELL), L'Hospitalet del Llobregat, 08908 Barcelona, Catalonia, Spain. [2] Department of Pediatrics, University of British Columbia, Vancouver, BC V6H 0B3, Canada. [3] Department of Biosciences, Faculty of Sciences and Technology (FCT), University of Vic – Central University of Catalonia (UVic-UCC), Vic, 08500 Barcelona, Catalonia, Spain. [4] Department of Pathology, University Hospital of Bellvitge, Oncobell, Bellvitge Institute for Biomedical Research (IDIBELL), L'Hospitalet del Llobregat, 08908 Barcelona, Catalonia, Spain. [5] Department of Biochemistry and Molecular Medicine, School of Medicine and Health Sciences, The George Washington University, Washington, DC 20037, USA. [6] Hereditary Cancer Program, Catalan Institute of Oncology, Girona Biomedical Research Institute (IDIBGI), 17190 Girona, Catalonia, Spain. [7] Terry Fox Laboratory, British Columbia Cancer Agency, Vancouver, BC V5Z 1L3, Canada. [8] Princess Margaret Cancer Research Centre, University Health Network, Toronto, ON M5G 2M9, Canada. [9] Department of Medical Biophysics, Faculty of Medicine, University Health Network, University of Toronto, Toronto, ON M5G 1L7, Canada. [10] Department of Pathology, University of British Columbia, Vancouver, BC V6T 1Z7, Canada. [11] Michael Cuccione Childhood Cancer Research Program, British Columbia Children's Hospital, Vancouver, BC V6H 3N1, Canada. [12] Hereditary Cancer Program, Catalan Institute of Oncology, Oncobell, Bellvitge Institute for Biomedical Research (IDIBELL), L'Hospitalet del Llobregat, 08908 Barcelona, Catalonia, Spain. [13] Biomedical Research Network Centre in Cancer (CIBERONC), Instituto de Salud Carlos III, 28222 Madrid, Spain. [14] Biomedical Research Network Centre in Respiratory Diseases (CIBERES), Instituto de Salud Carlos III, 28222 Madrid, Spain. [15]These authors contributed equally: Francesca Mateo, Zhengcheng He, Lin Mei. ✉email: cmaxwell@bcchr.ubc.ca; mapujana@iconcologia.net

Breast cancer is the most common malignancy in women worldwide[1]. Family history of the disease is a strong risk factor and about 15–20% of the familial risk is explained by inherited, rare pathological variants in the breast cancer 1 and 2 genes (*BRCA1* and *BRCA2*, respectively)[2,3]. Pathological variants of *BRCA1* confer a cumulative risk of breast cancer of 40–87% by 70 years of age[4,5]. This cancer type (hereafter *BRCA1*-associated breast cancer) is often diagnosed as having a triple-negative phenotype, which is characterized by a low or null level of expression of estrogen receptor α (ERα), progesterone receptor, and epidermal growth factor receptor 2, and/or a basal-like molecular subtype, expressing basal cell markers[6]. Tumors with these features generally present aggressive histopathological characteristics and are associated with relatively poor patient outcome[6].

The variable penetrance of pathogenic variants in *BRCA1* and *BRCA2* is partially due to environmental, lifestyle, and individual biological factors[7]. Dozens of common genetic variants can modify breast cancer risk in these settings, each of them having a relatively small effect[8]. Pooled analysis of genetic modifiers offers the opportunity of improving risk estimation and prevention[9]. The hyaluronan-mediated motility receptor (HMMR, also known as RHAMM) interacts with BRCA1 to regulate cell division and apicobasal polarization of mammary epithelial cells[10,11], and is a potential modifier of *BRCA1*-associated breast cancer risk[11,12]. However, the molecular and cellular alterations produced by the risk modifiers are yet to be identified, and the significance of the initial carcinogenic alterations in shaping features that define disease subtypes and progression is uncertain.

In this work, we analyze human germline and tumor data, and perform molecular and cellular studies in mouse models with conditional overexpression of HMMR and loss of BRCA1 and p53 in mammary epithelial cells, to decipher the biological foundation of increased risk of *BRCA1*-associated breast cancer. Our results uncover the relevance of modifiers beyond risk estimation, exposing how a modifier, HMMR, substantially shapes cancer cell and tumor microenvironment features.

## Results

**Prediction of biological processes underlying *BRCA1*-associated breast cancer risk.** Many genetic variants have been shown to modify the risk of *BRCA1*-associated breast cancer[8,9], but it is unclear whether the identified variants converge on defined biological alterations. To predict processes that, when altered, underpin *BRCA1*-associated breast cancer, we analyzed a collection of gene sets ($n = 6,289$), representing diverse states of health and disease[13], from the results of a meta-analysis of genome-wide association studies (GWASs) of *BRCA1*-associated and triple-negative breast cancer[14,15]. Using a scoring algorithm that limits false-positive results[16], 673 (10.7%), 200 (3.2%), and 25 (0.4%) sets were found to be associated with risk at consecutive significance thresholds (chi-squared test $p < 0.05$, <0.01, and 0.001, respectively; Fig. 1a and Supplementary Data 1a–c). Mining cancer-related keywords in the published abstracts describing the identified sets suggested diversity in the underlying biology of risk (Fig. 1b and Supplementary Data 1d, e). Examination of keyword co-occurrences highlighted the highest frequencies of pairs of terms including "cancer, damage, development, immune, mutation, proliferation, signaling, stem, stress, and/or transcription" (Fig. 1c and Supplementary Data 2a–c).

The risk-linked gene sets included four associated with *BRCA1* expression profiles or mutational status (Supplementary Data 1a). To assess this connection further, we computed the expression scores of the 673 identified sets across primary breast tumors of The Cancer Genome Atlas (TCGA)[17], and determined their correlations with *BRCA1* and genes coding for functional interactors of BRCA1[18]. The resulting 673 sets-*BRCA1*/BRCA1-interactor correlations were found to be significantly greater than those of 673 sets-random genes (Fig. 1d and Supplementary Data 3a, b). This suggests that diverse biological alterations underlie *BRCA1*-associated breast cancer risk, but a substantial fraction of this basis could be linked to *BRCA1* and BRCA1-interactor profiles.

HMMR is functionally connected to BRCA1[10]. BRCA1 and HMMR interact to regulate microtubule structures involved in the correct apicobasal polarization of mammary epithelial cells[11]. Common genetic variation in *HMMR* was proposed to be associated with breast cancer risk in carriers of pathological variants in *BRCA1*, but not in *BRCA2*[11,12]. This association was identified from the results of the variant rs299290 (T > C; minor allele frequency = 0.25), which was not significant at the genome-wide level, but nominally significant in two study phases[12,14,19] (combined log-relative risk effect = 0.044, $p = 0.001$). *BRCA1* and *HMMR* were identified in 49 (7.3%) and 17 (2.5%) of the 673 risk-linked gene sets, respectively, and six sets were found to be shared (hypergeometric test $p = 7.3 \times 10^{-4}$; Supplementary Data 4). We then assessed the expression correlation between *HMMR* and each of the 673 sets across TCGA primary tumors, and detected a subset of positive coefficients (Fig. 1e and Supplementary Data 5a). The correlations between *HMMR* and the 673 sets were significantly different between tumors originating from germline *BRCA1* pathological variants and luminal A (ERα-positive) tumors (Fig. 1f and Supplementary Data 5b). Therefore, *HMMR* expression in breast tumors correlates with gene sets predicted to influence *BRCA1*-associated breast cancer risk. However, the modifier effect of *HMMR* is not understood, but might include biological processes predicted by the risk-linked gene sets.

**Relative overexpression of *HMMR* may underpin a *BRCA1*-modifier effect.** The *HMMR* rs299290 variant corresponds to a predicted non-deleterious Val368Ala change (Ensembl protein ENSP00000351554). Examination of data from the Genotype-Tissue Expression (GTEx)[20] project indicated that rs299290 is an expression-quantitative locus (eQTL) for *HMMR*. The rs299290 minor allele (C) is associated with *HMMR* overexpression in several tissue types, including normal breast (Supplementary Fig. 1a). Analysis of TCGA breast cancer data confirmed that rs299290 is an *HMMR* eQTL (Fig. 2a). Intriguingly, although the eQTL effect was found to be similar across breast cancer subtypes (Supplementary Fig. 1b), basal-like tumors harboring the rs299290-TC and rs299290-CC genotypes were found to be specifically associated with poorer prognosis (Fig. 2b and Supplementary Fig. 2). Pan-cancer analysis further established rs299290 as an *HMMR* eQTL and revealed the rs299290-CC genotype to be frequently associated with features of genomic instability (Supplementary Fig. 3). *HMMR* expression in breast tumors was also found to be positively correlated with risk-linked gene sets involved in the DNA damage response (Supplementary Data 5a).

**Conditional overexpression of *HMMR* in mouse mammary epithelium increases *Brca1*-mutant tumorigenesis.** The cumulative evidence presented above suggested that *HMMR* overexpression increases *BRCA1*-associated breast cancer risk by further perturbing foundational process(es) that are altered in this type of cancer. To test this, we generated a Cre/*loxP*-dependent mouse model with the human *HMMR* full-length coding sequence cloned downstream from a *loxP*-STOP-*loxP* cassette in the *Rosa26* locus. To induce *HMMR* expression, we

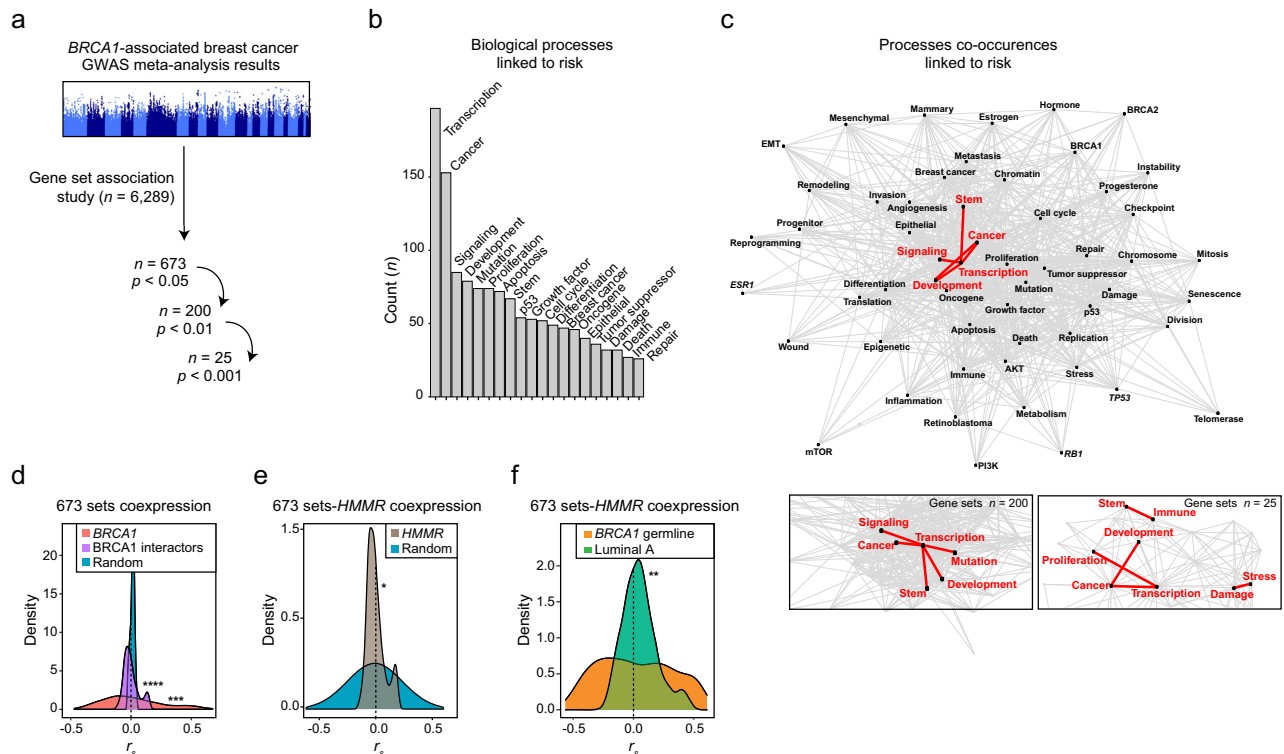

**Fig. 1 Curated gene sets linked to *BRCA1*-associated breast cancer risk and expression correlation with candidate modifier *HMMR*. a** Gene set-based analysis of the summary statistics of the *BRCA1*-associated and triple-negative breast cancer GWASs. The number of gene sets analyzed and of those found to be significantly associated at three scoring thresholds are indicated. **b** Biological annotation of 673 risk-linked gene sets using text mining. The 20 most frequently identified keywords in the corresponding publication abstracts are shown (*n*, counts). **c** Top panel, network of the keywords mined across published studies of the 673 risk-linked gene sets. Centrality is proportional to the number of instances found, and edge length is inversely proportional to the number of times that two given keywords appear in a given abstract. Red edges and keywords depict the five most frequent keyword pairs. Bottom panels, zoom-in into the most frequent pairs considering the subsets of 200 and 25 risk-linked gene sets. **d** Distributions of the coexpression coefficients (Spearman's correlation coefficient, $r_s$) between the 673 sets and *BRCA1*, genes coding for BRCA1 interactors, or equivalent randomly chosen genes (x1000), across TCGA primary breast tumors. The asterisks indicate significant difference relative to random (Wilcoxon test; ***$p = 0.001$ and ****$p < 0.0001$). **e** Distribution of coexpression coefficients ($r_s$) between the 673 risk-linked gene sets and *HMMR*, and equivalent null distribution across TCGA primary breast tumors. The asterisk indicates significant difference: two-tailed Student's paired-samples *t* test; *$p = 0.010$; *HMMR* 95% confidence interval (CI) 0.007–0.055; 671 degrees of freedom. **f** Distributions of coexpression coefficients ($r_s$) between *HMMR* and the 673 risk-linked gene sets across primary tumors with germline *BRCA1* pathological variants (*n* = 18; indicated "*BRCA1* germline") or luminal A tumors (*n* = 234). The asterisks indicate significant difference (Wilcoxon test; **$p = 0.002$).

chose to use the β-lactoglobulin promoter-*Cre* (*Blg-Cre*) transgene. Maximum allele recombination by *Blg*-driven Cre occurs after two pregnancies[21] and, importantly, *Blg*-Cre-driven loss of *Brca1* in mammary cells gives rise to tumors that phenocopy *BRCA1*-associated breast cancer[22]. Recombination of the *loxP*-STOP-*loxP* cassette was confirmed by targeted polymerase chain reactions (PCRs) in DNA extracted from mammary tissue of monoallelic and biallelic *HMMR* transgenic (*HMMR*^Tg/+ and *HMMR*^Tg/Tg, respectively) parous mice, and was not observed in matched liver tissue samples (Supplementary Fig. 4a). Transduction of a Cre-expressing vector in primary fibroblast from these mice demonstrated transgene- and dose-dependent over-expression of human HMMR (Supplementary Fig. 4b). The level of HMMR expression mediated by the transgene was lower than that observed in MCF7 (ERα-positive) breast cancer cells (Supplementary Fig. 4b). Similar Cre-transduction assays in primary cultures of mouse mammary epithelial cells (MECs) also over-expressed HMMR, as quantified by immunofluorescence analyses (Supplementary Fig. 4c). However, *Blg*-Cre-driven HMMR overexpression in mammary glands did not prompt tumor development or cause atypical epithelial structures to appear (Supplementary Fig. 5) in virgin and parous mice followed for at least 47 weeks: *Blg-Cre;HMMR*^Tg/+, virgin *n* = 20 and parous *n* = 24; *Blg-Cre;HMMR*^Tg/Tg, virgin *n* = 25 and parous *n* = 25; and including controls *HMMR*^Tg/+ parous *n* = 3, and *HMMR*^Tg/Tg parous *n* = 3.

To test the predicted *BRCA1*-modifier effect by *HMMR* overexpression, mice carrying the *HMMR* transgene were crossed with *Blg-Cre;Brca1*^f/f(exons 22–24)*;Trp53*^+/− mice[21,22] and the following mixed-background parous groups were monitored for mammary tumorigenesis: *Blg-Cre;HMMR*^Tg/Tg*;Brca1*^f/f*;Trp53*^+/− *n* = 25; *Blg-Cre;HMMR*^Tg/+*;Brca1*^f/f*;Trp53*^+/− *n* = 25; and *Blg-Cre;Brca1*^f/f*;Trp53*^+/− *n* = 27. Mice were genotyped using PCRs (Supplementary Fig. 6), and the ectopic expression of *HMMR*, and downregulation of *Brca1* and *Trp53* expression in developed tumors relative to control tissue were confirmed by semi-quantitative reverse-transcriptase PCR (qRT-PCR) assays (Supplementary Fig. 7).

Consistent with the original studies[21,22], parous *Blg-Cre;Brca1*^f/f*;Trp53*^+/− mice showed an incidence of mammary tumor of 59.3% (16/27) with a mean latency of 36.8 weeks (95% CI, 31.8–41.7 weeks). Notably, induction of one *HMMR* transgene allele increased the tumor incidence to 72.0% (18/25) and reduced latency to 29.5 weeks (95% CI, 25.6–33.4 weeks); and induction of the two *HMMR* transgene alleles further increased tumor incidence to 80.0% (20/25) and reduced latency

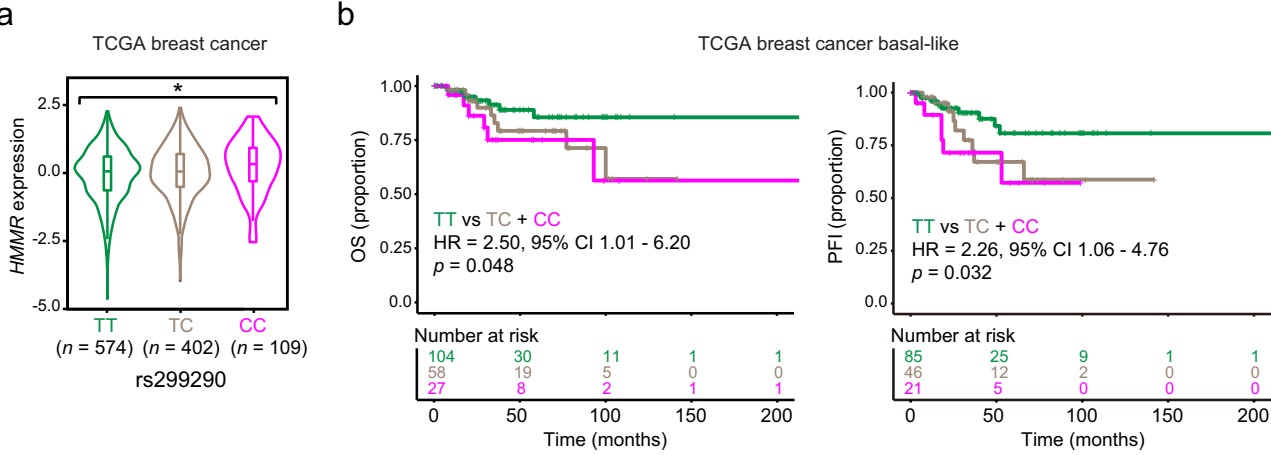

**Fig. 2 rs299290 is a breast cancer *HMMR* eQTL and is associated with the outcome of basal-like cases. a** Violin plot of *HMMR* expression in TCGA primary breast tumors according to rs299290 genotype (color-coded). In the box plots inside violin plots the horizontal lines represent the sample medians, the boxes extend from first to third quartile, and the whiskers indicate values at 1.5 times the interquartile range. The number of tumors of each genotype is indicated. One-way ANOVA; *$p = 0.033$. **b** Kaplan–Meier plots showing the association between overall survival (OS; left panel) or progression-free interval (PFI; right panel) and rs299290 in basal-like breast cancer. The hazard ratio (HR), 95% CI, and log-rank *p* value between cases with rs299290-TT and TC + CC genotypes are shown. The number of patients at risk at each time point are also indicated.

to 25.5 weeks (95% CI, 21.3–29.8 weeks; log-rank $p = 0.0038$; Fig. 3a). The effect size of one and two transgene alleles of *HMMR* were: hazard ratio (HR) = 2.32 (95% CI, 1.13–4.75; $p = 0.021$) and 3.34 (95% CI, 1.66–6.70; $p = 0.0007$), respectively. Including the covariates of tumor metaplasia, mitotic rate, and keratin 8 and/or 14 positivity (subsequent section), the $HMMR^{Tg/+}$ and $HMMR^{Tg/Tg}$ effect estimates on *Brca1*-mutant tumorigenesis yielded similar results: HR = 2.32 (95% CI, 0.97–5.53; $p = 0.058$) and 4.46 (95% CI, 1.72–11.56; $p = 0.002$), respectively. Probe-based quantification showed approximately 1.3-fold and 2.4-fold overexpression of human *HMMR* relative to endogenous *Hmmr* in developed tumors with the $HMMR^{Tg/+}$ and $HMMR^{Tg/Tg}$ genotypes, respectively, and about 1.9-fold and 2.1-fold overexpression in the analogous settings of mammary glands prior to tumor detection (Supplementary Fig. 8). These data indicate that a relatively modest rise in the level of HMMR expression in mammary epithelial cells increases the penetrance of *Brca1*-mutant mammary tumors, which parallels the association of the rs299290-C eQTL with an increased risk of *BRCA1*-associated breast cancer.

**Conditional overexpression of *HMMR* in mouse mammary epithelium influences *Brca1*-mutant tumor features.** The tumors that originated in the three surveyed groups of *Brca1*-mutant mice (Fig. 3a) were commonly classified as high grade and several showed metaplastic features, as expected[21] (Supplementary Fig. 9 and Supplementary Data 6). All analyzed tumors were found to be ERα-negative, negative for or weakly expressing keratin 8, and keratin 14-positive (Supplementary Fig. 10 and Supplementary Data 6), similar to breast tumors that tend to arise in carriers of *BRCA1* pathological variants. Next, four $HMMR^{Tg/Tg}$;$Brca1^{f/f}$;$Trp53^{+/−}$ and four $Brca1^{f/f}$;$Trp53^{+/−}$ tumors with a similar average time at diagnosis (31 and 32 weeks post-induction, respectively) were subjected to RNA sequencing (RNA-seq), and the transcriptome profiles of the two groups compared. Analysis of Gene Ontology (GO) terms showed that tumors with the $HMMR^{Tg/Tg}$ genotype overexpressed genes linked to angiogenesis and the immune system, among other related biological processes (Fig. 3b). Gene set-based analysis identified "TNF signaling via NF-κB", which corresponded to gene targets of NF-κB downstream of TNF signaling[23], as the most overexpressed set in

$HMMR^{Tg/Tg}$ tumors (Fig. 3c and Supplementary Table 1a). The overexpression of interleukin-6 and interleukin-10 (*Il6/10*) genes in this set, and of the angiogenic factor *Vefga*, was verified by qRT-PCRs (Fig. 3d). These observations extended the initial GWAS-predictions of immune-related gene sets associated with *BRCA1*-associated breast cancer risk, which included the identification of a set corresponding to TNF targets upon radiation[24] (Supplementary Data 1a).

Analysis of gene sets significantly underexpressed in $HMMR^{Tg/Tg}$;$Brca1^{f/f}$;$Trp53^{+/−}$ relative to $Brca1^{f/f}$;$Trp53^{+/−}$ tumors identified the "Reactome tight junction interactions" pathway (Fig. 3e and Supplementary Table 1b). Immunohistochemical assays confirmed underexpression of the claudin-1/3 (CLDN1/3) tight junction proteins in tumors with the $HMMR^{Tg/Tg}$ genotype (Fig. 3f). In turn, these tumors overexpressed the mesenchymal markers *Slug*, *Twist1*, and *Vim* (Fig. 3g). Then, quantification of tumor vessels and immune cell infiltration by CD31 and CD45 staining, respectively, revealed significantly higher contents in $HMMR^{Tg/Tg}$;$Brca1^{f/f}$;$Trp53^{+/−}$ than in $Brca1^{f/f}$;$Trp53^{+/−}$ tumors (Fig. 3h, i). Thus, tumors with the $HMMR^{Tg/Tg}$ genotype display features of basal-like and claudin-low breast cancers, such as ERα negativity, high levels of angiogenesis, TNF-NF-κB signaling, immune cell infiltration, and epithelial-to-mesenchymal (EMT) characteristics[25]. Indeed, these murine tumors displayed gene expression signatures that were positively associated with the immunomodulatory and mesenchymal-like classes of human triple-negative breast cancers[26] (Supplementary Fig. 11), implicating cross-species dysregulation of foundational pathways that modify the tumor microenvironment and cancer-cell phenotype.

**Overexpression of *HMMR* increases *Brca1*-mutant genomic instability that connects to mesenchymal and inflammatory features.** To further decipher the cellular changes mediated by *HMMR* overexpression, we studied MECs dissociated from tissues taken from 6-week-old virgin *Blg-Cre;$HMMR^{Tg/Tg}$;$Brca1^{f/f}$;$Trp53^{+/−}$* and *Blg-Cre;$Brca1^{f/f}$;$Trp53^{+/−}$* mice. As Cre-mediated recombination events are absent at this time, the cells were transduced with lentivirus expressing EGFP-only or EGFP-Cre, sorted 24 h post-transduction, and their phenotypic alterations assessed in colony-forming cell (CFC) assays. Cre-driven HMMR overexpression and BRCA1 underexpression were confirmed by immunofluorescence

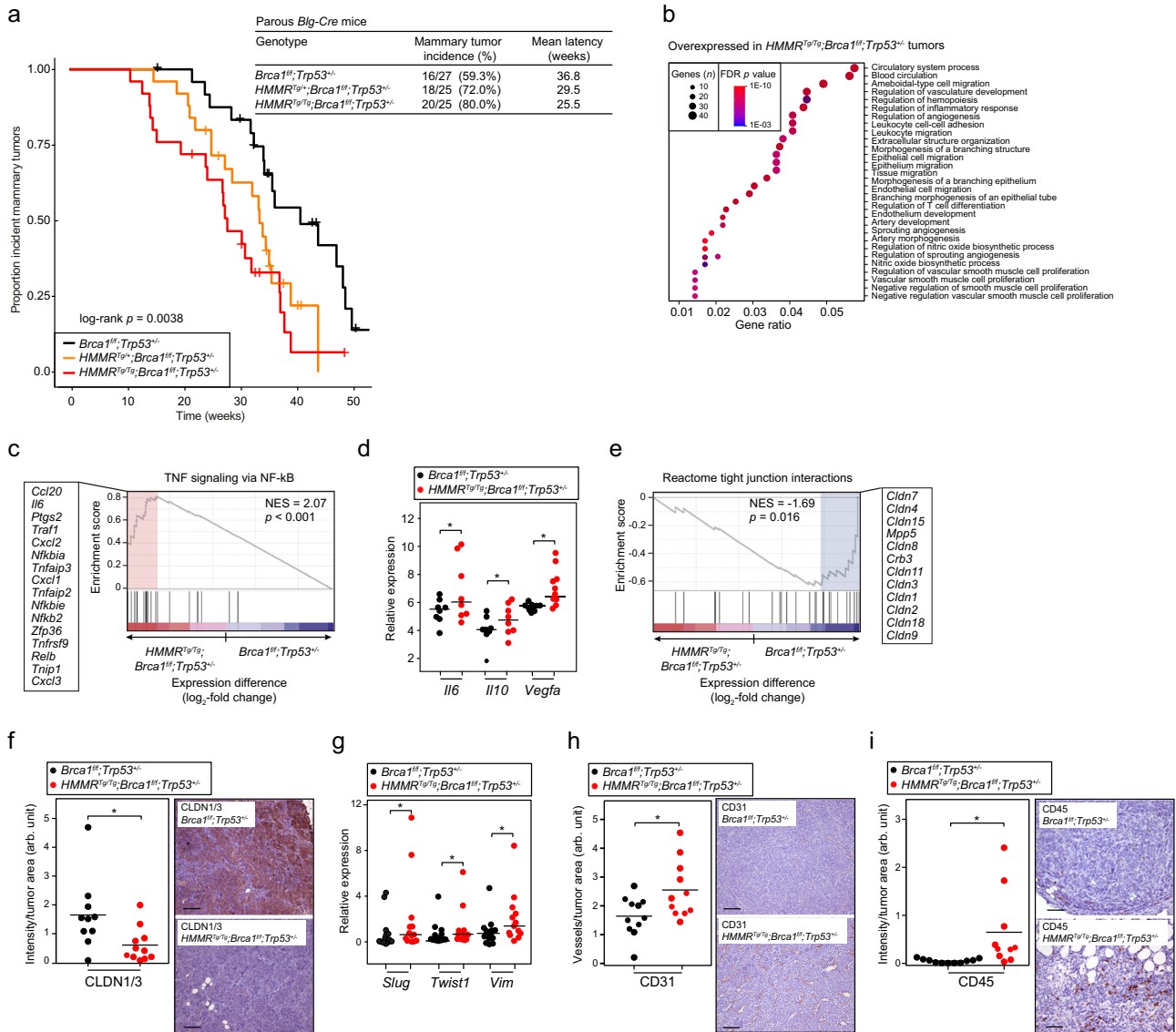

**Fig. 3 *HMMR* overexpression increases *Brca1*-mutant mammary tumorigenesis, and modifies the cancer cell phenotype and tumor microenvironment.**
**a** Left panel, Kaplan–Meier curves of mammary tumor incidence in mice divided into three genotype groups that differ by *HMMR* transgene status, as depicted in the inset. All mice were carriers of *Blg-Cre*, whose expression was induced by two rounds of pregnancy. Right top panel, table with details of the number of animals studied of each genotype, incidence of tumors, and mean latency. **b** Overrepresented (false discovery rate (FDR) < 1%; gene set size and FDR value color-coded) GO terms in genes overexpressed in *HMMR^{Tg/Tg}*;*Brca1^{f/f}*;*Trp53^{+/−}* relative to *Brca1^{f/f}*;*Trp53^{+/−}* tumors. **c** Gene Set Enrichment Analysis (GSEA) results showing overexpression of the "TNF signaling via NF-κB" gene set in *HMMR^{Tg/Tg}*;*Brca1^{f/f}*;*Trp53^{+/−}* tumors. GSEA was applied adopting standard parameters and using the pre-ranked expression differences (log₂-fold change) between the two tumor sets; the normalized enrichment score (NES) and associated probability (Kolmogorov-Smirnov statistic and x1000 permutation test) are shown. The red rectangle indicates the overexpressed leading edge, and the corresponding gene names are depicted. **d** Overexpression of *Il6*, *Il10*, and *Vegfa* in *HMMR^{Tg/Tg}*;*Brca1^{f/f}*;*Trp53^{+/−}* tumors (inset, color-coded) corroborated by semi-qRT-PCR assays. One-tailed Student's unpaired-samples *t* test; *\*p* = 0.035 (*Il6*), 0.045 (*Il10*), and 0.013 (*Vegfa*). Horizontal lines depict the average tumor values (*n* = 2 experiments). **e** GSEA results showing underexpression (Kolmogorov-Smirnov statistic and x1000 permutation test) of genes coding for tight-junction proteins (Reactome pathway) in *HMMR^{Tg/Tg}*;*Brca1^{f/f}*;*Trp53^{+/−}* tumors. **f** Quantification of CLDN1/3 expression by immunohistochemistry assays and representative images of tumor staining in defined groups. Arbitrary units (arb. unit). One-tailed Student's unpaired-samples *t* test; *\*p* = 0.011. Scale bar = 100 μm. **g** Overexpression of *Slug*, *Twist1*, and *Vim* in *HMMR^{Tg/Tg}*;*Brca1^{f/f}*;*Trp53^{+/−}* tumors. One-tailed Student's unpaired-samples *t* test; *\*p* = 0.045 (*Slug*), 0.033 (*Twist1*), and 0.045 (*Vim*). Horizontal lines depict the average tumor values (*n* = 2 experiments). **h**, **i** Quantification (arb. units) of CD31 and CD45 by immunohistochemistry assays, and representative images of tumor staining in defined groups. One-tailed Student's unpaired-samples *t* test; *\*p* = 0.013 (CD31) and *\*p* = 0.011 (CD45). Scale bar = 100 μm.

analysis on day 5 (Supplementary Fig. 4c, d). The CFC assays generally produced colonies with epithelial characteristics, comprised of densely packed cells expressing CLDN1 and zonula occludens (ZO-1) markers, but sparse, vimentin (VIM)-positive colonies with EMT features could also be identified (Supplementary Fig. 12a–c). We observed that colony output was significantly

reduced, but EMT colonies were more frequently found in both CFC-assay genotypes incorporating Cre expression (Fig. 4a). In addition, we observed that Cre expression produced colonies composed of cells with larger, relatively more variably shaped nuclei, and that HMMR overexpression significantly exacerbated these nuclear alterations (Fig. 4b and Supplementary Fig. 12d). Detailed

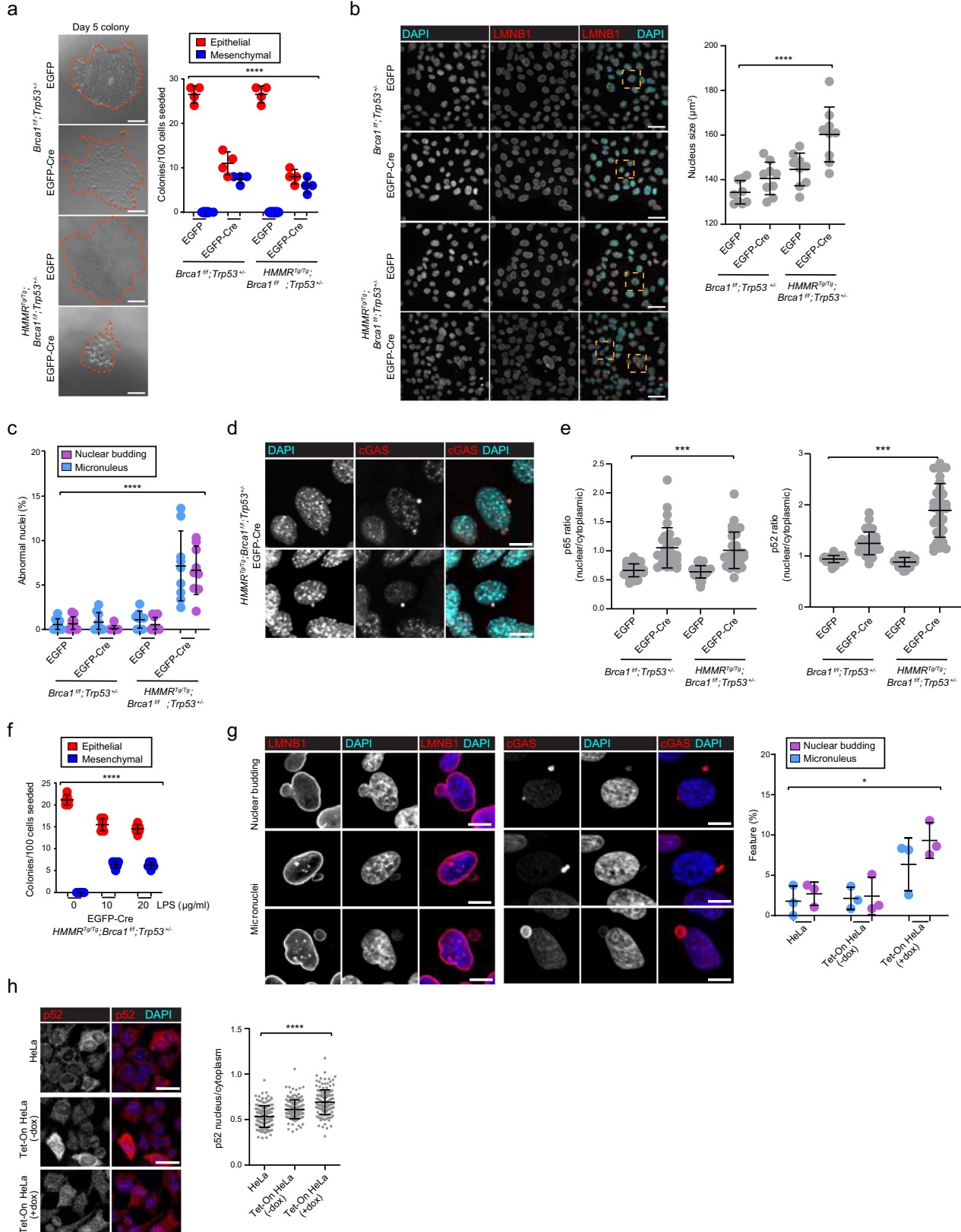

examination of the nuclear architecture showed that colonies with HMMR overexpression also had more frequent nuclear budding and micronuclei (Fig. 4c and Supplementary Fig. 12d), which were found to localize the cyclic GMP-AMP synthase (cGAS) (Fig. 4d), a cytosolic sensor of double-stranded DNA that fosters inflammatory gene expression[27–30]. This signaling was also linked to

BRCA1-associated breast cancer risk with the identification of cGAS-stimulator of interferon genes (STING) and innate immune system-related gene sets in the analysis of the GWAS results (Supplementary Data 1a).

Activation of cGAS-STING by genomic instability and micronuclei[27–30] can trigger canonical and non-canonical NF-κB

**Fig. 4 BRCA1 loss and HMMR overexpression alters the epithelial cell phenotype and nuclear structure, prompting genomic instability and p52 nuclear expression. a** Left panel, bright-field images of day 5 CFC derived from *Blg-Cre;Brca1^{f/f};Trp53^{+/−}* and *Blg-Cre;HMMR^{Tg/Tg};Brca1^{f/f};Trp53^{+/−}* MECs transduced with EGFP-empty or EGFP-Cre lentivirus. Scale bar = 100 μm. Right panel, colony output and colony phenotype in day 5 CFC (inset, color-coded; mean ± standard error of the mean (s.e.m.) from $n = 4$ wells; $n = 2$ experiments; $n = 100$ cells/well). One-way ANOVA; ****$p < 0.0001$. **b** Left panels, lamin B1 (LMNB1) detection and/or 4′,6-diamidino-2-phenylindole (DAPI) showing nuclei and nuclear lamina architecture in subconfluent MEC cultures. Scale bar = 40 μm. Right panels, quantification of nucleus size (μm^2; mean ± standard deviation (s.d.) from $n = 9$ frames; $n = 3$ experiments; >125 cells/experiment). One-way ANOVA; ****$p < 0.0001$. **c** Frequency of micronucleus and nuclear blebbing in day 5 colonies (mean ± s.e.m. from $n = 6$ wells; $n = 3$ experiments; >125 cells/experiment). One-way ANOVA; ****$p < 0.0001$. **d** cGAS-positive micronuclei in *Blg-Cre;HMMR^{Tg/Tg};Brca1^{f/f};Trp53^{+/−}* MECs transduced with EGFP-Cre lentivirus. Scale bar = 10 μm. **e** Ratio (nucleus/cytoplasm) of p65 and p52 intensity in day 5 colonies (mean ± s.d. from $n = 30$ cells; $n = 2$ experiments). One-way ANOVA; ***$p = 0.001$. **f** Colony phenotype of *Blg-Cre;HMMR^{Tg/Tg};Brca1^{f/f};Trp53^{+/−}* EGFP-empty MECs treated with LPS during colony formation (mean ± s.e.m. from $n = 6$ wells; $n = 3$ experiments; 100 cells seeded per well). One-way ANOVA; ***$p = 0.001$. **g** Nucleus budding and micronuclei (LMNB1, left panel), and micronuclei (cGAS, middle panel) detection in HeLa cells overexpressing HMMR. Scale bar = 10 μm. Right panel, percentage of cells displaying nucleus budding or micronuclei (mean ± s.d. from $n = 3$ experiments with 105-242-253 (HeLa); 151-183-208 (Tet-On -dox); and 147-155-176 (Tet-On +dox) cells). One-way ANOVA; *$p = 0.014$. **h** Left panels, p52 nuclear/cytoplasm localization in analyzed cell settings. Scale bar = 30 μm. Right panel, p52 nuclear/cytoplasm ratio (mean ± s.d. from $n = 3$ experiments consisting of 150 (HeLa), 150 (-dox), and 150 (+dox) cells). One-way ANOVA; ****$p < 0.0001$.

signaling[31,32], and promote EMT[31] and tumorigenesis[33]. High levels of activity of these pathways have been identified in basal-like and claudin-low breast cancers[34]. We observed that loss of BRCA1 in non-transformed MECs activated both NF-κB pathways, as measured by nuclear p65/RELA and p52/NF-κB2 staining, respectively, and HMMR overexpression further boosted non-canonical NF-κB signaling (Fig. 4e and Supplementary Fig. 13). Analysis of gene sets corresponding to curated RELB:p52 and RELA:p50 targets revealed significant overexpression of the former set in *HMMR^{Tg/Tg};Brca1^{f/f};Trp53^{+/−}* tumors (Supplementary Fig. 14). We then tested whether activation of NF-κB signaling was sufficient to favor an EMT phenotype in EGFP-transduced (control) *Blg-Cre;HMMR^{Tg/Tg};Brca1^{f/f};Trp53^{+/−}* MECs. Induction of EMT was clear in colonies exposed to lipopolysaccharide (LPS) and this transition occurred without reduced colony output (Fig. 4f). Therefore, HMMR overexpression in MECs augments *Brca1*-mutant genomic instability, and this alteration may foster EMT and non-canonical NF-κB signaling, which are also initially promoted by *Brca1* mutation.

**Overexpression of *HMMR* disrupts cortical localization of ARPC2.** Micronuclei can originate from abnormal cell division[35]. To precisely assess the impact of HMMR overexpression on cell division, we used HeLa cells engineered to express GFP-HMMR upon exposure to doxycycline (dox, Tet-On system)[36] (Supplementary Fig. 15a). Doxycycline exposure roughly doubled HMMR expression, which resulted in a significant delay in metaphase progression and mitotic spindles that frequently misoriented with respect to the dividing cell's long axis (Supplementary Fig. 15a–d). Similar to Cre-transduced *HMMR^{Tg/Tg};Brca1^{f/f};Trp53^{+/−}* MECs, HeLa cells overexpressing HMMR showed a substantial increase in the frequency of nucleus budding and cGAS-positive micronuclei (Fig. 4g), and of the level of activation of NF-κB signaling, as detected by p52 staining (Fig. 4h). Imaging mitosis and progeny cells mechanistically connected HMMR overexpression with extensive blebbing during anaphase, and asymmetrical cell sizes and genome instability in resultant daughter cells (Supplementary Fig. 16). Consistent with this, pronounced membrane elongation events were commonly observed during anaphase (Fig. 5a), and daughter cell sizes were frequently unequal in HeLa cells overexpressing HMMR (Fig. 5b). Therefore, HMMR overexpression disrupts mitotic cortex integrity, which causes micronucleation in the progeny cells.

Immunoprecipitation-mass spectrometry (IP-MS) assays targeting HMMR in M-phase synchronized HeLa cells were performed to discover mitotic mechanisms related to cell cortex

stability. Proteins known to interact with HMMR, including CALML5, CHICA, and DYNLL1, were identified in these assays (Supplementary Fig. 17). The assays also suggested the existence of HMMR complexes containing actin-binding proteins, including ACTR3 —a major constituent of the ARP2/3 complex— and the non-muscle myosins MYH10 and MYO18A (Supplementary Fig. 17). Actomyosin contractility near spindle poles protects against aneuploidy[37], and ARP2/3 localization at the periphery of cells contributes to mitotic cortical stability[38]. Therefore, we analyzed the cortical localization of ARP2/3, through its subunit ARPC2, and MYH10 and related MYH09 during anaphase in HeLa cells overexpressing HMMR. The localization of non-muscle myosins was not altered (Supplementary Fig. 18), but the degree of cortical retention of the ARP2/3-member ARPC2 was significantly lower in anaphase cells overexpressing HMMR (Fig. 5c–e). ARP2/3 tunes centrosome microtubules during anaphase[39], and ARPC2 transiently located to pericentrin-positive anaphase spindle poles, as expected (Supplementary Fig. 19a–c); however, overexpression of HMMR prematurely recruited ARPC2 to the spindle in prometaphase (Supplementary Fig. 19d, e) and, in anaphase, ARPC2 colocalized with HMMR-decorated mitotic spindles, concurrently with the less frequent localization in the cortex (Fig. 5e–g). Analogously, ARPC2 cortical localization was diminished, and spindle pole localization was boosted in Cre-expressing *HMMR^{Tg/Tg};Brca1^{f/f};Trp53^{+/−}* MECs (Fig. 5h–j).

**Disruption of ARPC2 cortical localization is mediated by AURKA-HMMR.** HMMR binds to microtubule and actin filaments in vitro and in cells[40], and published evidence recognizes four interactors shared between HMMR and ARPC2 (AURKA, LRRK2, NDC80, and PRKACA)[18]. Endogenous co-immunoprecipitation assays of HMMR in HeLa cells did not detect an association with the major constituent of the ARP2/3 complex, ACTR3 (Supplementary Fig. 20), which suggests that a putative interaction may be indirect and/or spatiotemporally restricted at the mitotic spindle poles. Analysis of ARPC2 localization during mitosis confirmed restricted recruitment to the spindle poles during metaphase-anaphase, with HMMR overexpression prompting earlier recruitment, at prophase-prometaphase (Supplementary Figs. 19 and 21). Then, to test for a mechanism linking HMMR overexpression and ARPC2 localization at the spindle poles, we examined the consequences of inhibiting mitotic kinases whose activity is known to be promoted by HMMR expression: AURKA[41,42], CSNK1A1[43], and PLK1[36]. Inhibition of AURKA with 1 nM MLN8237 was able to normalize both mitotic blebbing and daughter cell size in HeLa cells overexpressing HMMR, but these phenotypic normalizations were not observed with

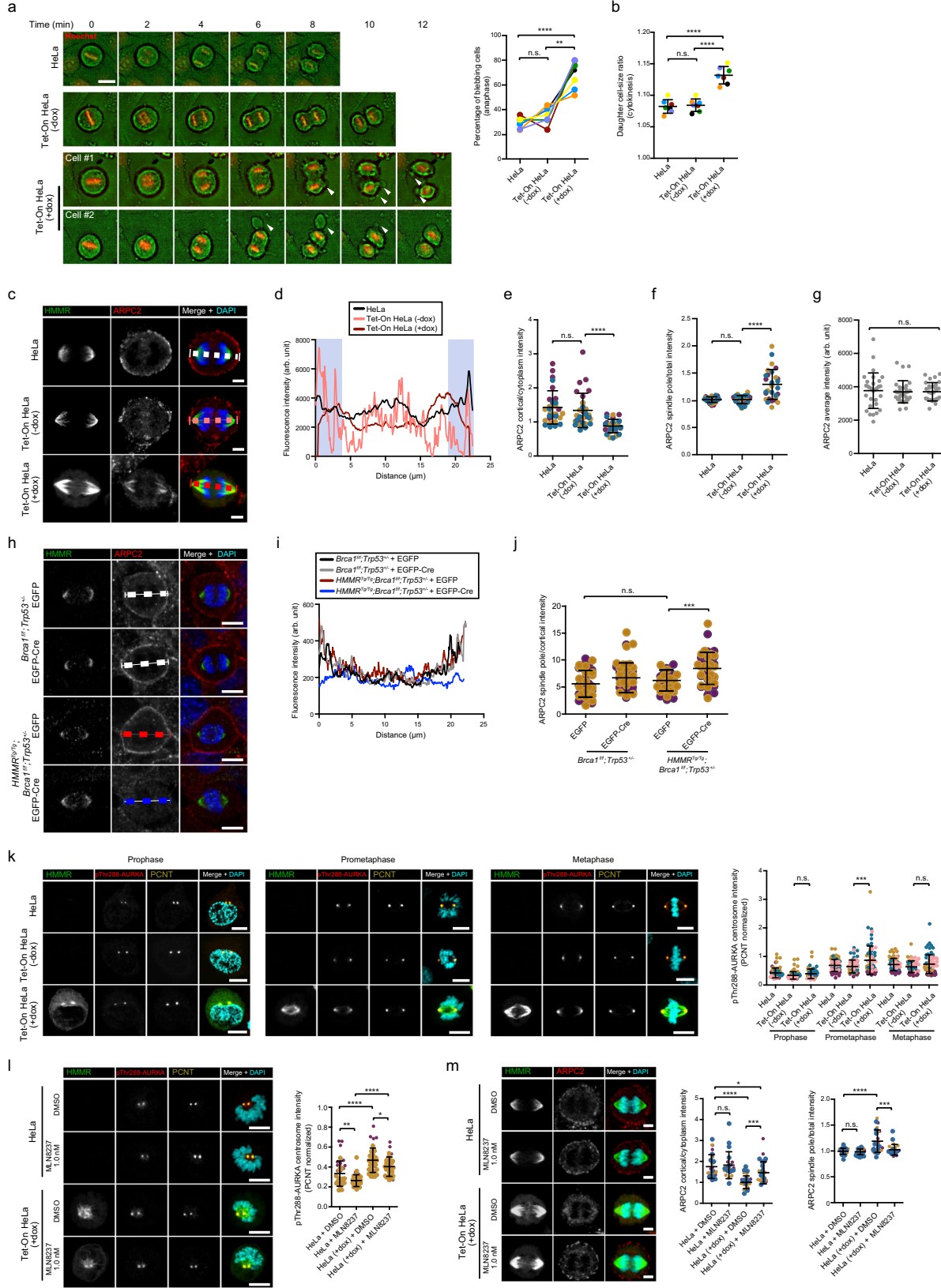

inhibition of CSNK1A1 or PLK1 (Supplementary Fig. 22). In addition, evaluation of AURKA activity, as measured by pThr288-AURKA signal intensity at mitotic pericentrin (PCTN)-positive centrosomes, revealed overactivation in HeLa cells overexpressing HMMR (Fig. 5k). Moreover, exposure to 1 nM MLN8237 reduced pThr288-AURKA intensity (Fig. 5l), and reduced spindle pole

localization, while normalized cortical localization of ARPC2 (Fig. 5m). These observations are consistent with frequent over-expression of AURKA and HMMR in *BRCA1*-mutant breast cancer cell lines and tumors[10,11], and suggest that abnormal earlier localization of ARPC2 at mitotic spindle poles is mediated by overexpression of HMMR that triggers AURKA overactivation.

**Fig. 5 HMMR overexpression decreases ARPC2 cortical localization and activates AURKA. a** Left panel, mitotic time-lapse analysis of parental and Tet-On (-dox or +dox) HeLa (arrowheads indicate membrane blebs). Scale bar = 20 μm. Right panel, quantification of anaphase blebs (mean ± s.d.; $n = 7$ experiments (color-coded)). Two-tailed Student's paired (**$p = 0.002$) or unpaired (****$p < 0.0001$) $t$ test. Not significant (n.s.). **b** Daughter cell size ratio (mean ± s.d.; $n = 7$ experiments; $n = 175$ cells/setting). Two-tailed Student's paired-sample $t$ test; ****$p < 0.0001$. **c** HMMR and ARPC2 detection in anaphase HeLa cells. Color-coded dashed lines indicate the measurement of the corresponding profiles in subsequent panel. Scale bar = 5 μm. **d** Intensity profile (arb. units) across dashed lines. The blue-shaded areas mark 3 μm distance from the cortex. **e** ARPC2 cortical/cytoplasm intensity in anaphase (blue-shaded/non-shaded ratio; mean ± s.d.; $n = 3$ experiments; $n = 30$ cells/setting, likewise in (**f**) and (**g**)). Two-tailed Student's paired-samples $t$ test; ****$p < 0.0001$. **f** ARPC2 spindle pole/total in anaphase (integrated intensity/area). **g** ARPC2 total intensity (arb. units) in anaphase. **h** HMMR and ARPC2 detection in MECs. Scale bar = 10 μm. **i** Anaphase intensity (arb. units) across dashed lines in (**h**). **j** ARPC2 spindle pole/cortical ratio in MECs (mean ± s.d.; $n = 2$ experiments; $n = 40$ cells/setting). Two-tailed Student's paired-samples $t$ test; ***$p = 0.0001$. **k** HMMR, pThr288-AURKA, and PCTN detection in HeLa settings. Right panel, pThr288-AURKA centrosomal intensity (mean ± s.d.; $n = 4$ experiments). Two-tailed Student's paired-samples $t$ test; ***$p = 0.0001$. Scale bar = 10 μm. **l** Left panels, markers detected in HeLa exposed to DMSO or AURKA inhibitor. Right panel, pThr288-AURKA intensity differences (mean ± s.d.; $n = 2$ experiments). Two-tailed Student's paired-samples $t$ test; *$p = 0.036$, **$p = 0.006$, and ****$p < 0.0001$. **m** Left panels, HMMR and ARPC2 detection in HeLa exposed to DMSO or AURKA inhibitor. Middle-right panels, ARPC2 cortical/cytoplasm and spindle pole/total ratios (mean ± s.d.; $n = 3$ experiments). Two-tailed Student's paired-samples $t$ test; *$p = 0.021$, ***$p = 0.0004$ (cortical), ***$p = 0.0007$ (spindle), and ****$p < 0.0001$.

**Altered cell division and immune microenvironment changes in premalignant tissue**. To assess the depicted alterations in the premalignant mammary state, contralateral tissue was collected at the time of incident tumors in *Blg-Cre;HMMR*$^{Tg/Tg}$*;Brca1*$^{f/f}$*;Trp53*$^{+/-}$ and *Blg-Cre;Brca1*$^{f/f}$*;Trp53*$^{+/-}$ mice ($n = 6$ per parous group; average age at sacrifice of 45.6 and 49.3 weeks, respectively). Elevated expression of HMMR was confirmed in *Blg-Cre; HMMR*$^{Tg/Tg}$*;Brca1*$^{f/f}$*;Trp53*$^{+/-}$ tissue (Supplementary Fig. 23). Then, RNA-seq analysis revealed overexpression of gene sets involved in cell division and chromosome segregation (Supplementary Fig. 24a), and underexpression of actin binding-related gene sets (Supplementary Fig. 24b) in *Blg-Cre;HMMR*$^{Tg/Tg}$*;Brca1*$^{f/f}$*; Trp53*$^{+/-}$ tissue. Mitotic cells were significantly more frequent in this setting (Supplementary Fig. 25a), but there was not a coincident increase in CCNB1-positive cells (Supplementary Fig. 25b), suggesting that delayed progression through mitosis may account for the increased mitotic figures. Importantly, the cortical localization of ARPC2 was significantly reduced in mitotic cells (Fig. 6a) and frequent alteration in nuclear size, elevated levels of cGAS-positive micronuclei, and elevated p52 expression were also observed in the epithelial cells of *Blg-Cre;HMMR*$^{Tg/Tg}$*;Brca1*$^{f/f}$*; Trp53*$^{+/-}$ tissue (Fig. 6b–d), which are each consistent with the ex vivo studies of Cre-transduced primary MECs.

A recent study of single-cell RNA-seq profiles across premalignant mammary stages in *Blg-Cre;Brca1*$^{f/f}$*;Trp53*$^{+/-}$ mice has revealed pro-tumorigenic cell and tissue microenvironment changes[44], reminiscent of our observation of enhanced immune cell infiltration in tumors with the *HMMR*$^{Tg/Tg}$ genotype. Analyzing these profiles, we found significant overexpression of endogenous *Hmmr* through premalignant stages (Supplementary Fig. 26a). Moreover, the pathways of "TNF signaling via NF-κB" and "KEGG cytosolic DNA sensing" were also found to be overexpressed in this setting (Supplementary Fig. 26b). In turn, "Reactome tight junction interactions" significantly decreased during the premalignant stages, including in luminal progenitors (Supplementary Fig. 26b, c).

Having observed the single-cell RNA-seq profiles, we then analyzed the expression of immune response genes in ex vivo assays of MECs from *Blg-Cre;HMMR*$^{Tg/Tg}$*;Brca1*$^{f/f}$*;Trp53*$^{+/-}$ and *Blg-Cre;Brca1*$^{f/f}$*;Trp53*$^{+/-}$ mice. The MECs overexpressing HMMR showed a higher level of expression of genes of the NF-κB and TNF pathways, as well as of macrophage colony-stimulating factors, *Csf1-3* (Supplementary Fig. 27a). To assess the degree to which cGAS was responsible for producing these differences, the *Blg-Cre;HMMR*$^{Tg/Tg}$*;Brca1*$^{f/f}$*;Trp53*$^{+/-}$ MECs were treated with a cGAS inhibitor (0.7 μM RU.521) or vehicle and the expression of six proinflammatory genes was determined:

inhibition of cGAS caused significant downregulation of four of the examined genes (*Csf1, Il1a, Nfkb1*, and *Nfkb2*; Supplementary Fig. 27b). CSFs correlate with tumor-associated macrophage (TAM) density and breast cancer mortality[45]. We found a higher level of infiltration of CD45-positive cells in premalignant *Blg-Cre;HMMR*$^{Tg/Tg}$*;Brca1*$^{f/f}$*;Trp53*$^{+/-}$ tissue (Fig. 6e). These cells were characterized by the expression of the TAM markers F4/80, CD68, and VCAM1 (F4/80$^{+}$CD68$^{+}$VCAM1$^{+}$; Fig. 6f). Therefore, HMMR overexpression exacerbates protumorigenic changes to the immune microenvironment that are correlated with lower latency and greater penetrance of mammary tumorigenesis.

**Protumorigenic features in cancer-of-origin cells**. The premalignant gene expression profiles observed in *Blg-Cre;Brca1*$^{f/f}$*; Trp53*$^{+/-}$ mice further suggested that *HMMR* overexpression amplifies perturbations that are present at the origin of *BRCA1*-associated breast cancer. We then found that *HMMR* expression is positively correlated with the "TNF signaling via NF-κB" and "KEGG cytosolic DNA sensing" pathways in human basal-like and/or claudin-low tumors, but not in the other major breast cancer subtypes (Fig. 7a). We subsequently examined the human cell type of origin of *BRCA1*-associated breast cancer for key protein expression or localization differences. This study included primary luminal progenitors isolated from breast tissue of healthy women non-carriers ($n = 3$) and carriers of *BRCA1* pathological variants ($n = 3$). Overexpression of HMMR and abnormal ARPC2 spindle/cortical localization in mitotic cells, and increases in nuclear localization of p52 were each identified in luminal progenitors from carriers of *BRCA1* pathological variants (Fig. 7b). These results were also found to be consistent with frequent overexpression of HMMR in *BRCA1*-mutant breast cancer cell lines and tumors[10,11], and with impaired formation of MCF10A acini upon concurrent depletion of BRCA1 and HMMR[11]. These findings establish a link between BRCA1 loss and HMMR overexpression in promoting breast carcinogenesis.

**Discussion**
From mouse to human analyses, this study reveals a sequence of molecular, cellular, and tissue microenvironment alterations that is associated with increased breast cancer risk (Fig. 7c). The consequences of *HMMR* overexpression against a *Brca1*-mutant background expose the initial interplay of these alterations that may promote development of *BRCA1*-associated breast cancer. Analysis of GWAS results indicates that a substantial fraction of risk modification is mediated by perturbation of established BRCA1 functions and/or that of its interactors[46], including

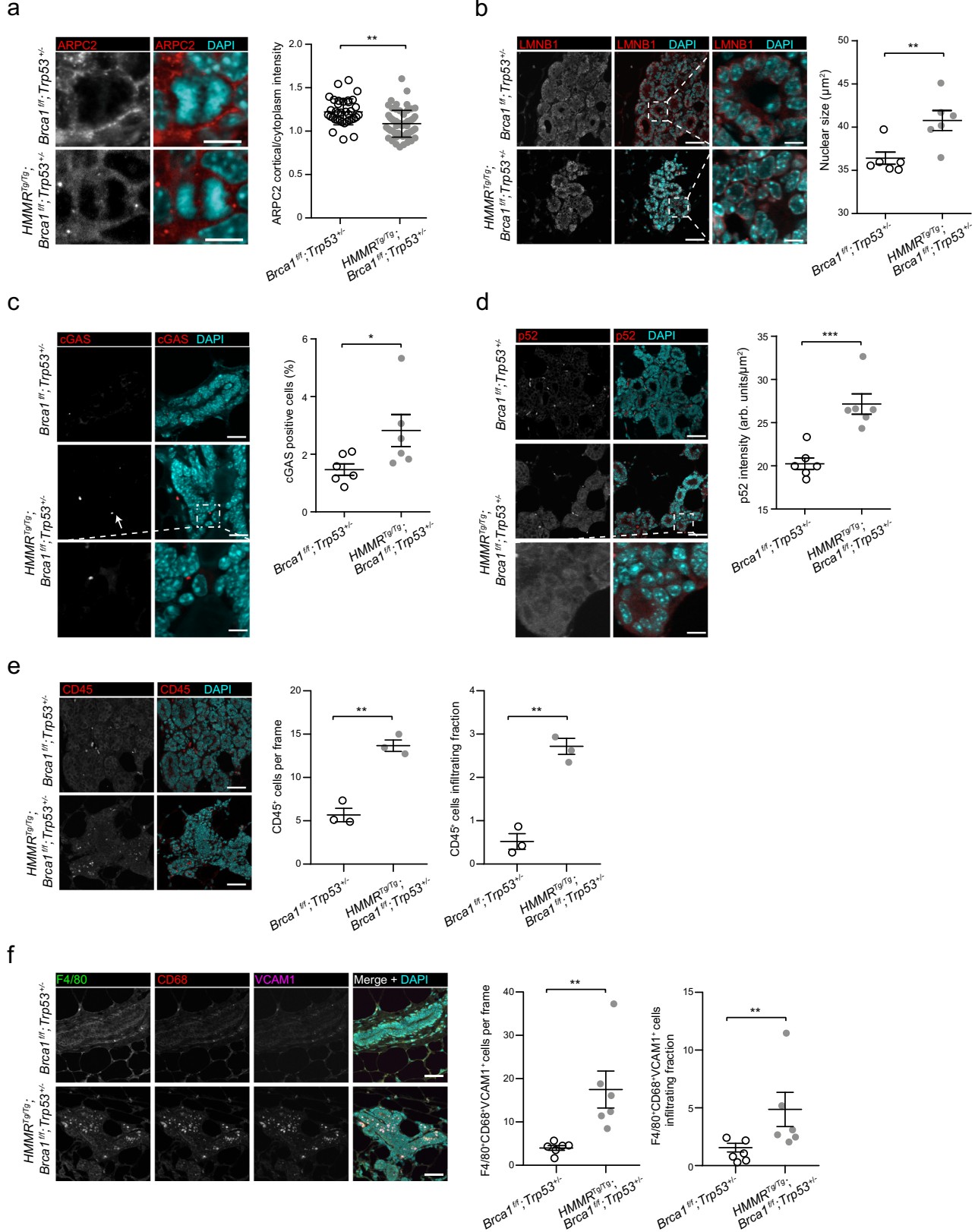

HMMR[10,11]. This analysis also predicted involvement of immune system processes, which may then interact with increased genomic instability. Thus, the results of our study lead us to propose that perturbation of cell division due to loss of BRCA1 and overexpression of HMMR, which in turn enhances AURKA activity that diminishes mitotic cortex stability mediated by ARPC2, increases genomic instability in daughter cells. Genomic instability may then trigger cGAS-STING signaling, which could subsequently activate inflammatory signals through NF-κB. Interestingly, *BRCA1*-associated breast cancers show active NF-κB-driven transcriptional programs[47], and BRCA1-deficiency causes persistent NF-κB signaling[48]. Boosted

**Fig. 6 Detection of HMMR-mediated perturbations in premalignant mouse mammary tissue. a** Cortical enrichment of ARPC2 in premalignant mammary epithelial cells. Mitotic cells (mean ± s.d.; $n = 40$ cells for $HMMR^{Tg/Tg}$;$Brca1^{fl/fl}$;$Trp53^{+/−}$ + EGFP from six tissue; mean ± s.d.; $n = 60$ cells for $HMMR^{Tg/Tg}$; $Brca1^{fl/fl}$;$Trp53^{+/−}$ + EGFP-Cre from four tissue) were analyzed in six mice per genotype. Scale bar = 10 μm. Two-tailed Student's unpaired-samples $t$ test; **$p = 0.002$. **b** Immunofluorescence analysis and quantitation of size of nuclei (μm$^2$) in premalignant mammary epithelial cells from six mice per genotype, using LMNB1 staining (mean ± s.e.m.). Two-tailed Student's unpaired-samples $t$ test; **$p = 0.009$. Scale bar = 20 μm and 4 μm (zoom). **c** Immunofluorescence analysis and quantitation of cGAS positive staining in tissue from six mice per genotype (mean ± s.e.m.). Two-tailed Student's unpaired-samples $t$ test; *$p = 0.046$. Scale bar = 20 μm and 4 μm (zoom). **d** Immunofluorescence analysis and quantitation of p52 in premalignant mammary epithelial cells from six mice per genotype (arb. units/μm$^2$; mean ± s.e.m.). Two-tailed Student's unpaired-samples $t$ test; ***$p = 0.0005$. Scale bar = 20 μm and 4 μm (zoom). **e** Immunofluorescence analysis and quantitation of CD45-positive (CD45$^+$) cells in premalignant mammary tissue from three mice per genotype. CD45$^+$ cells and those infiltrating the epithelial structures were measured per frame (mean ± s.e.m.; $n = 3$ frames per tissue; each frame >100 cells; $n = 3$ experiments). Two-tailed Student's unpaired-samples $t$ test; **$p = 0.001$ (per frame and infiltrating). Scale bar = 20 μm. **f** Immunofluorescence and quantitation of F4/80$^+$CD68$^+$VCAM1$^+$ macrophages in premalignant mammary tissue from six mice per genotype (mean ± s.e.m.; $n = 10$ frames per tissue; each frame >100 cells; $n = 2$ experiments). Two-tailed Student's unpaired-samples $t$ test; **$p = 0.002$ (per frame) and **$p = 0.009$ (infiltrating). Scale bar = 20 μm.

NF-κB signaling could promote EMT and additional signals that ultimately provide a permissive tumorigenic microenvironment by recruiting TAMs.

The conclusions from our study may be limited by several considerations. The modifier effect of other biological processes predicted to be connected to *BRCA1* and *HMMR* profiles remains to be assessed. *HMMR* rs299290 could also influence expression of neighboring genes and, thereby, combined deregulation of HMMR and other gene products might further affect the risk outcome and tumor features. The original variant was also shown to be associated with risk of breast cancer in Ashkenazi Jewish individuals who were non-carriers of *BRCA1* pathological variants, but not in non-Ashkenazi populations[10,49], which suggests there may be additional genetic differences in the *HMMR* locus that influence risk. In contrast to the proposed model, activation of cGAS-STING signaling could lead to cellular senescence and tumor suppression by promoting immunosurveillance[50,51]. The strength and duration of cGAS-STING activation, possibly combined with other molecular and cell alterations, may define the outcome towards tumor promotion or suppression[52]. The interplay between HMMR overexpression and loss of p53 remains to be examined, but it is expected to be connected to loss of BRCA1, since most *BRCA1*-associated breast cancers show *TP53* mutations[53]. Also, the proposed *BRCA1*-associated tumorigenic model is mainly based on correlations between genetic, molecular, and cellular perturbations, which, on the other hand, could be influenced by the genetic background and/or other confounding factors in which the studies were performed.

The proposed model parallels described immune cell changes and a luminal progenitor to basal/mesenchymal transition in *BRCA1*-associated breast tissue[44,54,55]. Interestingly, a cell-autonomous STING-driven inflammation and proangiogenic status has also been shown in *BRCA1*-deficient ovarian cancer cells[56]. In addition, some of the identified proinflammatory molecules were found to be induced by aging-associated alterations of mammary epithelial and stromal cells[57]. Moreover, our HMMR IP-MS assays identified the S100A7-9 proteins as candidate interactors, and these proteins modulate immune system homeostasis and inflammatory responses[58]. The S100 protein family are calcium-binding cytosolic proteins, and HMMR was previously shown to bind calmodulin in a calcium-dependent manner[40] and may interact with calcium-binding calmodulin-like 5 (CALML5). Further studies may be warranted to evaluate these potential interactions in modulating breast cancer risk and development.

Our model draws attention to the relevance of modifiers beyond their relatively small effects on risk estimates. Modifiers may also emerge as players influencing cancer cell and tumor microenvironment features, which could partially determine

disease progression and response to therapy[59]. At the level of cancer prevention, the observed upregulation of inflammatory factors highlights unexplored opportunities for targeted therapeutic approaches.

## Methods

**GWAS data and gene set analysis.** The GWAS summary statistics of *BRCA1*-associated and triple-negative breast cancer were downloaded from the public repository of the Consortium of Investigators of Modifiers of BRCA1/2 (CIMBA: http://cimba.ccge.medschl.cam.ac.uk/projects/). This study did not require individual data. The pathway scoring algorithm (Pascal)[16] was applied with default parameters considering genetic variants with minor allele frequency > 5% and using the gene-locus sum chi-squared test for significance assessment. All variants linked to a dbSNP 151 identifier were included. The gene sets corresponded to the curated collection of The Molecular Signatures Database (MSigDB version 7.4: http://www.gsea-msigdb.org/gsea/msigdb/). A weighted score was computed for each pair of defined keywords found in the publication abstracts of 395 gene sets (from 673 sets, 278 were not linked to publications). The inverse of all keyword pairs found in a given abstract was computed, and the total weighted score of a given pair was defined as the sum of all partial weights across the 395 abstracts. The scores were used to define edge length in undirected keyword networks constructed using the *ggnetwork*[60] (version 0.5.10) and *network*[61] (version 1.17.1) R packages.

**Rosa26 loxP-STOP-loxP-HMMR mouse model.** A mouse model with conditional expression of the human *HMMR* gene was generated by homologous recombination in embryonic stem (ES) cells (genOway's custom development). For this purpose, a targeting vector was constructed by cloning the human *HMMR* cDNA and hGHpA signal into the genOway Rosa26 Quick Knockin™ targeting vector. The construct included *HMMR* (full-length sequence verified; transcript NM_001142556, ENST00000393915) under the control of the endogenous *Rosa26* promoter and with a 5′-*loxP*-flanked STOP-neomycin selection (Neo) cassette that could be excised using Cre recombinase, subsequently activating transgene expression. The construct was linearized with the *Asc*I restriction enzyme and transfected into ES cells by electroporation. G418-resistant colonies were isolated and screened by PCR and Southern blot to test for homologous recombination at the 5′ end of the *Rosa26* locus and the 3′ end of the targeting vector. Several recombined ES cell clones were injected into C57BL/6J-recipient blastocysts, which were re-implanted into pseudo-pregnant females and allowed to develop to term. Resulting chimeric males were mated with C57BL/6 wild-type females and germline transmission of the knock-in allele was verified by PCR analysis of tail germline DNA from F1 mice. Heterozygous offspring were crossed to generate homozygous mice (F2). These mice were crossed with the *Brca1* mutant model corresponding to the *Trp53*^tm1Brd^ *Brca1*^tm1Aash/F22-24^ Tg(LGB-cre)74Acl/J mouse strain[21] (catalog 012620, The Jackson Laboratory), and the offspring with a mixed background evaluated for mammary tumor development. The *Blg-Cre* (LGB-cre) transgene is active during lactation and, to ensure expression of human *HMMR* and loss of *Brca1* in mammary tissue, the mice were allowed to go through two rounds of pregnancy and lactation and then set aside to develop tumors. Mice were monitored every 2–3 days for tumor development and euthanized when incident tumors were detected (tumor size < 1000 mm$^3$) or at the end of the study (no tumor detected). All animal experiments were carried out in the University of Barcelona-Bellvitge animal facility, under the Generalitat de Catalunya license authority (reference 9774) and with the permission of the IDIBELL University of Barcelona-Bellvitge Ethics Committee.

**Histology.** From each sacrificed mouse, a variety of tissues (mammary glands, liver, lung, and spleen) were freshly frozen or fixed in 4% formaldehyde in phosphate-buffered saline (PBS), embedded in paraffin, sectioned at 4 μm, routinely stained with hematoxylin and eosin, and microscopically examined by two

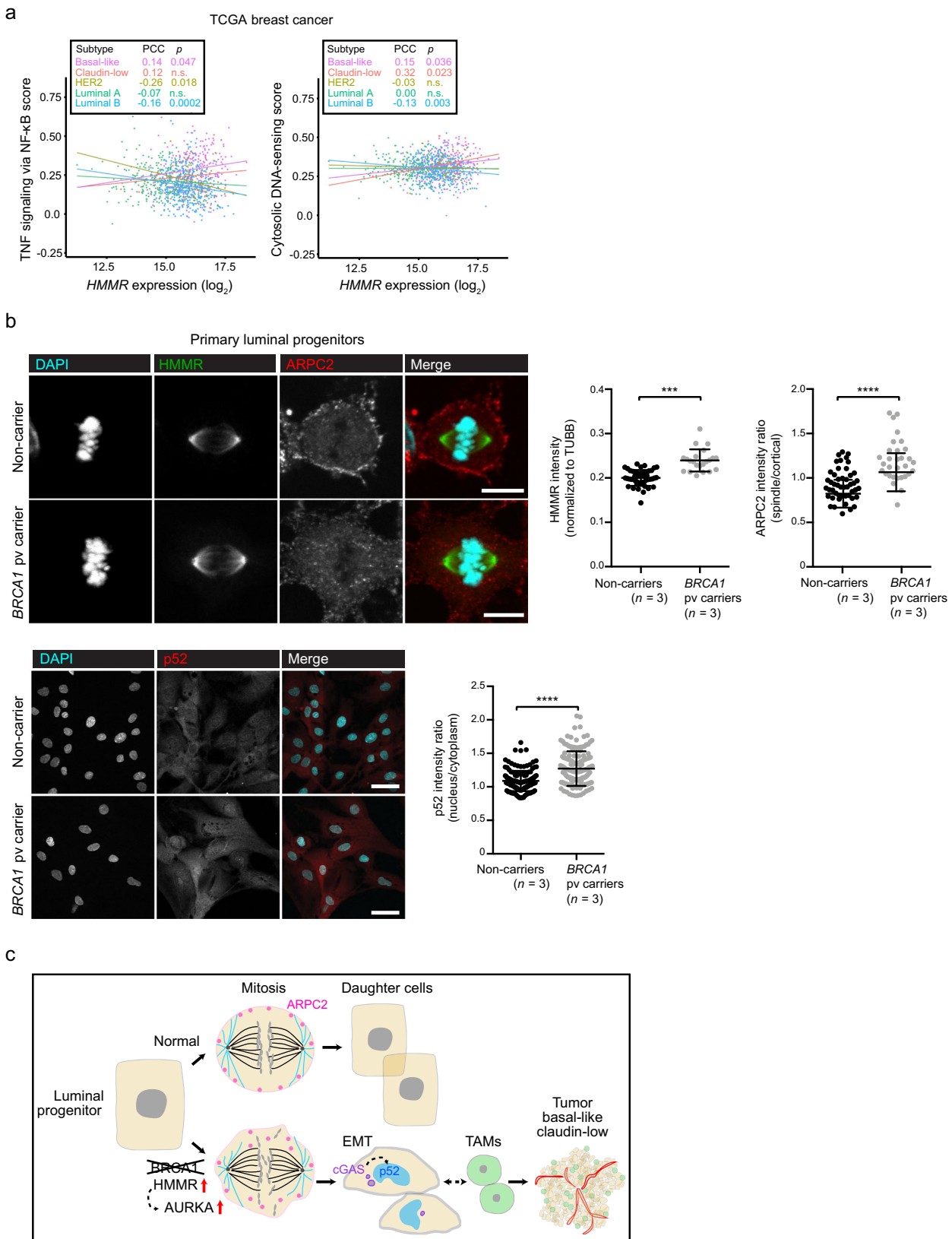

pathologists. Tumor features were scored blindly relative to their genotype, following established criteria for mouse studies[62].

**Gene expression analysis of mouse tumors**. The mRNAs from tumors were extracted using TRIzol (Thermo Fisher Scientific) and, following quality controls, single-end-sequenced at the IRB's facility in Barcelona. The RNA-seq reads were trimmed for adaptors, masked for low-complexity and low-quality sequences, and subsequently quantified for transcript expression using Kallisto[63] (version 0.43.16) and mouse genome version mm9. Gene-level quantification was carried out using the *tximport*[64] (version 3.14) Bioconductor package, mm9 and Ensembl v94 annotations. Differential expression was analyzed using DESeq2[65] (version 2.13. GO term enrichment was analyzed using clusterProfiler[66] (version 3.14) and GOnet[67] (version 2019-07-01). The pre-ranked GSEA[13] (version 4.1) method was

**Fig. 7 HMMR-centered perturbations at the basis of breast cancer risk. a** *HMMR* expression correlation (Pearson's correlation coefficient (PCC)) with the gene sets "TNF signaling via NF-κB" (left panel) and "Cytosolic DNA-sensing" (KEGG annotation; right panel) across breast cancer subtypes (inset, color-coded; basal-like, $n = 213$; claudin-low, $n = 51$; HER2, $n = 88$; luminal A, $n = 234$; and luminal B, $n = 556$). The data corresponded to TCGA primary breast tumor RNA-seq profiles and the PCC *p* values (or n.s.) are also indicated. **b** Top and bottom left panels, representative immunofluorescence images of ARPC2, HMMR, and p52 staining in luminal progenitors from healthy non-carrier and carriers of *BRCA1* pathological variants (pv). Right panels, quantification: ARPC2; mean ± s.d.; $n = 30$ cells from non-carriers ($n = 3$); and $n = 30$ cells from *BRCA1* pv carriers ($n = 3$); HMMR; mean ± s.d.; $n = 31$ cells from non-carriers; $n = 21$ cells from *BRCA1* pv carriers; and p52; mean ± s.d.; $n = 150$ cells in each setting. Two-tailed Student's unpaired-samples *t* test; ***$p = 0.0002$; ****$p < 0.0001$. Scale bars = 10 μm (ARPC2, HMMR) and 50 μm (p52). **c** Illustration of the proposed sequence of molecular and cellular events leading to normal luminal progenitor cell division (top sequence) and their alteration in *BRCA1*-associated breast cancer (bottom sequence). Loss of BRCA1 and HMMR overexpression activate AURKA and reduce cortical retention of ARPC2 during mitosis, disrupting correct chromosome segregation, prompting emergence of micronuclei in daughter cells, which appear imbalanced in size and have undergone EMT as a consequence. Then, micronuclei-induced cGAS signaling activates non-canonical NF-κB, which can facilitate recruitment of TAMs that enable initial tumorigenesis. As a consequence, developed tumors are predicted to show basal-like and claudin-low features with high vascularization and immune cell infiltration.

applied based on standard parameters. The RELB:p52 and RELA:p50 target sets were taken from the Harmonizome database[68]; the cytosolic DNA-sensing pathway gene set was taken from the KEGG database[69]; and the triple-negative breast cancer gene expression signatures were taken from the original publication[26].

For quantitative expression analyses, total RNA was isolated from tissue using TRIzol reagent (Thermo Fisher Scientific) and complementary DNA (cDNA) synthesized from 1 μg RNA using the High Capacity cDNA Reverse Transcription Kit (Thermo Fisher Scientific), following the manufacturer's protocol. The cDNA was diluted to a working concentration of 6 ng/μl with RNase-free water. Specific primer sets of oligonucleotides were designed, spanning exon/exon boundaries wherever possible. PowerUp SYBR Green PCR Master Mix (Applied Biosystems) was used for qPCR reactions. The PCR conditions used were as follows: 95 °C for 10 min, followed by 45 repeat cycles of 95 °C for 10 s, then 60 °C for 45 s. A dissociation stage of 95 °C for 15 s, 50 °C for 10 s, and 95 °C for 15 s was added. TaqMan probes were purchased from Thermo Fisher. TaqMan Universal PCR Master Mix (Applied Biosystems) was used in the reactions. The assay conditions were as follows: 95 °C for 10 min, followed by 45 repeat cycles of 95 °C for 15 s, and 60 °C for 1 min. Data generated from the qPCR reactions were analyzed using the 2-ΔΔCt method. All samples were run in triplicate and experiments repeated at least three times. The primer sequences and TaqMan™ assays used in this study are listed in Supplementary Table 2.

**TCGA and *Blg-Cre;Brca1f/f;Trp53+/−* public data.** TCGA data were obtained from the Genomic Data Commons Data Portal (https://portal.gdc.cancer.gov) and from the corresponding consortium publications. Individual genetic data were obtained following specific approval: dbGaP Data Access Committee project #11689. Gene expression data corresponded to FPKM-UQ values. The TCGA *BRCA1* germline and somatic mutation status was taken from a previous curation[70]. The gene set expression scores were computed using the single-sample GSEA (ssGSEA) algorithm calculated within the GSVA software[71] (version 1.43.1), and using primary breast tumor gene expression (RNA-seq FPKM-UQ) data. The bimodality of expression values was determined using the bimodality coefficient and Hartigan's dip statistic[72]. Preprocessed and normalized single-cell RNA-seq data of mammary premalignant and tumor stages of the *Blg-Cre;Brca1f/f;Trp53+/−* mouse model were obtained from the authors[44]. Only cells with at least 2,000 measured genes were used to compute ssGSEA scores.

**Immunohistochemistry and antibodies.** The assays were performed on serial paraffin sections using an EnVision kit (Dako). Antigens were retrieved using citrate-based (pH 6) buffer. Endogenous peroxidase was inactivated by pre-incubation in a solution of 3% $H_2O_2$, and blocked in 1x PBS with 10% serum. Slides were incubated overnight at 4 °C with primary antibody diluted in blocking solution. Secondary anti-mouse or anti-rabbit peroxidase-conjugated antibodies (Envision + system-HRP, Dako) or anti-rat (ImmPRES HRP, Vector Laboratories) were used. Sections were hematoxylin-counterstained and examined with a Nikon Eclipse 80i microscope. Images were captured under bright field and captured at 10X-40X magnification using a Nikon Digital Sight color video camera linked to a computer system. Quantifications were performed using Fiji-ImageJ (version 1.52s-10, National Institute of Health) software with the IHC profiler plugin. Color deconvolution was applied and DAB immunoreaction was selected based on the standard threshold. The quantified immunoreaction was measured per tumor area and represented as arbitrary units (arb. units). The antibodies used in this study are detailed in Supplementary Table 3.

**Isolation and culture of primary mouse mammary epithelial cells.** Highly purified MECs were isolated from cryopreserved mammary glands from 6-week-old mice. Briefly, this process involved dissociating viably cryopreserved preparations into single-cell suspensions. Cryopreserved mammary glands (around 1 $cm^3$) were cut into small pieces for better tissue dissociation, which was performed with gentleMACS (Miltenyi Biotec) following the manufacturer's protocols. After

obtaining a single-cell suspension, cells were cultured with EpiCult Plus (STEM-CELL Technologies) on collagen-coated plates (Corning) and the medium was refreshed every other day. For CFC assays, MECs were seeded at single-cell density (100 cells per 24 wells) in collagen-coated plates for 5 days, and LPS at different doses was used to activate NF-κB in MEC cultures.

**Cell culture and compounds.** HeLa cells were purchased from the American Type Culture Collection (ATCC, catalog CCL-2™) and cultured in 10% FBS DMEM, 20 U/ml penicillin (Invitrogen) and 20 μg/ml streptomycin (Invitrogen). Tetracycline-inducible (Tet-On) HeLa cells were created and cultured in DMEM with 10% tetracycline-free FBS (Clontech) plus additives, as described[73]. GFP-HMMR expression was induced with addition of 2 μg/ml doxycycline (Clontech). Expression of induced GFP-HMMR was visualized by fluorescence microscopy. Cell lines were grown at 37 °C in a 5% (v/v) $CO_2$ incubator. Cells were passaged at 85% density and reseeded at 15–20% density. The tested inhibitors against mitotic kinases (AURKA, CSNK1A1, and PLK1) are cytotoxic and impede progression through mitosis. To determine the optimal concentrations of each inhibitor for the screen, we first performed a dose response curve against HeLa cells using a cyto-toxicity assay (Abcam ab112118). For each inhibitor, HeLa cells were incubated with eight titrated doses, including a DMSO control. Curves were fitted using the dose-response-inhibition equation (log(inhibitor) - response) in Prism 6 software (GraphPad). Tet-On HeLa cells were induced to express EGFP-HMMR and then treated with sublethal doses of each inhibitor. Mitotic cells were followed by time-lapse microscopy after 2 h of incubation ($n = 3$ experiments). The frequency of anaphase blebbing and the daughter cell-size ratio at cytokinesis were analyzed as described. The 4T1 and MCF7 cell lines were obtained from ATCC (CRL-2539™ and HTB-22™, respectively) and were grown according to recommendations.

**Lentiviral production and transduction, and gene depletion.** HEK-293FT cells were obtained from Invitrogen (Thermo Fisher, catalog R70007) and maintained in 10% FBS DMEM at 37 °C, in a 5% $CO_2$ incubator and split at 70–80% confluence. Lentiviral particles and transduction were produced as previously described[74]. Lentivirus was produced by packaging EGFP (Addgene #36083) or EGFP-Cre (Addgene #86805) with packing plasmid psPAX2 (Addgene #12260) and envelope plasmid pMD2.G (Addgene #12259). MECs were seeded in 6-well plates at 80% confluence, and fresh medium was provided 4 h before transduction. MECs were incubated with the viral supernatants overnight and medium was changed in the morning. Protein expression was measured by immunofluorescence 72 h after transduction.

**Cell synchronization.** Mitotic HeLa cells were synchronized in M phase by treating cells with 200 ng/ml nocodazole (Sigma-Aldrich) for 16 h. Cells were then washed and incubated in a proteasome inhibitor, 15 μM MG132 (Sigma-Aldrich), for 3 h followed by collection. For monopolar synchronization, the cells were incubated with an Eg5 inhibitor (5 μM (+)-S-Trityl-L-cysteine (Sigma-Aldrich)) for 16 h and mitotic exit was forced by the addition of 20 μM RO-3306 (Sigma-Aldrich) for 5 min.

**Immunofluorescence and image acquisition.** Cells were seeded on coverslips and fixed with 4% paraformaldehyde at room temperature for 10 min, followed by cold methanol fixation at 4 °C for 10 min. Cells were permeabilized with PBS-0.25% Triton X-100 (Sigma-Aldrich) for 20 min. Fixed and permeabilized cells were washed with PBS three times and blocked in PBS with 0.1% Triton X-100 and 3% BSA for 1 h at room temperature. Antibodies were diluted in PBS with 0.1% Triton X-100 and 3% BSA accordingly. Coverslips were incubated in diluted antibodies for 2 h at room temperature, followed by three PBS washes. The coverslips were then incubated with diluted secondary antibodies at room temperature for 1.5 h. Cells were incubated in PBS-diluted Hoechst for 15 min, followed by two PBS washes and a d$H_2O$ wash prior to mounting. Coverslips were mounted with ProLong Gold Antifade Reagent (Invitrogen) and allowed to seal overnight at room temperature.

Fixed cells were imaged using Fluoview software (Olympus) connected to an Olympus Fluoview FV10i confocal microscope. Images were captured using a 60X-300 × 1–2 NA oil objectives as a stack of 5–7 optical sections with a spacing of 0.5 μm through the cell volume. The images were analyzed using Fiji-ImageJ (version 1.52s-10, National Institute of Health) to generate maximum intensity projection of the fluorescence channels.

**Live cell imaging for membrane blebbing analysis**. Cells were grown in plastic 96-well plates (Corning) and stained with Hoechst for 10 min, followed by three PBS washes before imaging with fresh media. Cells in the 96-well plates were placed in a 37 °C environmental chamber supplied with 5% $CO_2$ (ImageXpress Micro XL). The plates were imaged with the ImageXpress Micro XL epi-fluorescence microscope (Molecular Devices Incorporated) controlled by MetaXpress software (version 5.0.2.0. Molecular Devices Incorporated). Images were captured through a 40 × 0.75 NA dry objective with 2 × 2 binned resolution. For blebbing analysis, images were taken every 1 min. For F-actin-labeled live cell imaging of parental HeLa and Tet-On HeLa, the cells were incubated with Cell-Light Actin-RFP, BacMam 2.0 (Thermo Fisher Scientific, catalog C10583) 48 h before imaging, and Tet-On HeLa cells were treated with doxycycline 24 h before imaging. Images were captured every 5 min.

**Immunoprecipitation mass spectrometry**. HeLa cells were synchronized via exposure to 200 ng/ml nocodazole (Sigma-Aldrich) for 17 h, followed by a 2-h incubation with 15 μM MG132 (Sigma-Aldrich) to arrest at metaphase. Synchronized cells were then harvested and lysed in lysis buffer, as described[75], at 4 °C for 45 min. Cell lysates were pre-cleared by centrifugation and incubation with Protein A/g agarose beads (Sigma-Aldrich) for 30 min at 4 °C prior to incubation with antibodies. Pre-cleared lysates were either immunoprecipitated with anti-HMMR (Abcam ab124729) or rabbit IgG (Sigma-Aldrich 12–370) antibodies in separate tubes, and incubated with rotation for 24 h at 4 °C. The lysate-antibody mixture was collected with Protein A/g agarose beads (Sigma-Aldrich) and incubated at 4 °C for 22 h. The beads were then washed with lysis buffer (X3), wash buffer (X2), and PBS (X2) to remove detergent. Immunoprecipitated proteins were reduced with dithiothreitol (10 mM final concentration in 50 mM Hepes pH 8.5) and alkylated with iodoacetamide (40 mM final concentration in 50 mM Hepes pH 8.5), and digested with trypsin/Lys-C (Promega). Reductive dimethylation was used for stable isotope labeling of IgG control and HMMR immunoprecipitated samples. The conditions were combined, peptides purified on C18-STAGE tips, and separated over 125 min on a 50 cm C18 column connected to an Easy nLC ultra-high-pressure LC system (Thermo Fisher Scientific) coupled to a Q Exactive HF mass spectrometer. Acquired spectra were searched using Proteome Discoverer software (Thermo Fisher Scientific, version 2.5) against the *Homo sapiens* reference proteome including isoforms, downloaded from the UniProt database[76] (version 2017–07). The searches used a 1% false discovery rate-cut-off at peptide and protein level, and the following modifications were considered: static modifications, +57.021 Daltons (Da) on C residue; variable modifications, +15.995 Da on M, and +28.031 or +34.063 on K; and peptide N terminus, +42.011 Da. Differential expression in anti-HMMR versus rabbit IgG immunoprecipitates was determined using a Student's *t* test and proteins commonly identified in affinity enrichment experiments were flagged based on information derived from the CRAPome (version 2.0) database[77].

**Gene expression analysis of contralateral mammary tissue**. Three consecutive sections of paraffin-embedded contralateral tissue from each selected mouse with an incident tumor were used for RNA extraction using the AllPrep DNA/RNA FFPE kit (QIAGEN), and concentration determined using the Qubit 3.0 Fluorometer (Invitrogen). Library preparation was performed on an Ion Chef and Ion Torrent S5 platforms (Thermo Fisher Scientific) following Ion AmpliSeq™ Library Preparation on the Ion Chef™ System Quick Reference. The resulting cDNA library was quantified using the Ion Library TagMan™ Quantification Kit (catalog 4468802). Targeted sequencing was performed on an Ion Chef and Ion Torrent S5 platforms following manufacturer's protocols (Thermo Fisher Scientific). The Ion AmpliSeq Transcriptome Mouse Gene Expression Assays measure gene expression of over 20,000 mouse RefSeq genes in a single assay simultaneously. Basal data processing and quality control was performed using the AmpliSeqRNA plug-in for Ion Torrent S5. Differential gene expression analysis was performed in Transcriptome Analysis Console (version 4.0.1). Gene ontology terms for genes with differential expression identified from RNA-seq or TaqMan™ assays were identified with GOrilla[78] (version 01–2019) and g:Profiler[79] (version 0.2.1).

**Human breast tissue and luminal progenitors**. Highly purified subpopulations of human MECs were isolated from normal reduction mammoplasty tissue samples from randomly chosen premenopausal women non-carriers and carriers of *BRCA1* pathological variants (*n* = 3/group; mutations c.68_69del, p.Glu23fs; c.1687C > T, p.Gln563Ter; and c.66dupA, p.Glu23Argfs), as previously described[80,81]. The breast tissue were confirmed to be histologically normal. Briefly, cell population isolation involved dissociating viably cryopreserved organoid preparations into single cell suspensions, and then isolating viable EpCAM⁺CD49f⁺ luminal progenitors using cell-sorting gates that excluded hematopoietic (CD45⁺), endothelial (CD31⁺), dead (DAPI⁺) cells, and debris. All donors provided informed written

consent and the study was approved by the University of British Columbia's Ethics Review Board (reference H19-04034).

**Statistics**. Univariate survival analyses of rs299290 genotypes in TCGA breast cancer subtypes and of mammary mouse tumor incidence were performed using the Kaplan–Meier method and log-rank test, computed with the *survival* (version 3.2–13) R package. Multivariate Cox proportional regression models of the effect of *HMMR* in mouse mammary tumorigenesis included the covariates of metaplasia (yes/no), mitotic rate (%), and keratin 8 and/or 14 positivity (0–3 score). Tumor grade was not considered because most cases were classified as high grade. Other statistical analyses generally involved two-tailed Student's unpaired samples *t*-tests, or unpaired one-way ANOVAs followed by Tukey's multiple-comparison tests, the exception being that two-tailed Student's paired-samples *t*-tests were used to compare primary cell data, including CFC, multipolar mitosis, and mitotic outcomes. All differences were considered to be significant for values of $p < 0.05$. All molecular and cellular assays were repeated independently at least two times, found to show a similar trend. The number of experiments and replicates is indicated in each relevant figure legend.

**Reporting summary**. Further information on research design is available in the Nature Research Reporting Summary linked to this article.

## Data availability

The RNA-sequencing data generated in this study have been deposited in the Gene Expression Omnibus database under accession number GSE163756 (preneoplastic mammary tissue) and GSE164004 (mammary tumors). The raw and processed data IP-MS data generated in this study have been deposited in the Proteome Exchange (PXD031752) and MassIVE (MSV000088870: https://doi.org/10.25345/C57659F06) repositories. The publicly available GWAS data used in this study are available in the CIMBA consortium web page (https://cimba.ccge.medschl.cam.ac.uk/projects/) and the publicly available single-cell *Brca1*-associated mouse mammary tumorigenesis data used in this study are available in the ArrayExpress database under accession code E-MTAB-10043 (https://www.ebi.ac.uk/arrayexpress/experiments/E-MTAB-10043/; processed data is also available in http://marionilab.cruk.cam.ac.uk/BRCA1Tumourigenesis). The remaining data are available within the Article, Supplementary Information or Source Data file. Source data are provided with this paper.

## Code availability

The R-code used to compute the weighted score of literature keywords and to construct undirected networks is available at GitHub (https://github.com/pujana-lab/HMMR).

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

## Acknowledgements

We thank members of the ICO's Bioinformatics Unit for their encouragement with this study. We also wish to thank the BCAC and CIMBA consortia for making publicly available the summary statistics of GWASs (CIMBA acknowledgments are detailed in a supplementary file); Professor Matt Smalley for initial guidance on the mouse studies; and the University of Barcelona–Bellvitge Animal Facility for supporting these studies. Our results are partly based upon data generated by the TCGA Research Network (https://www.cancer.gov/tcga), and we are grateful to the TCGA consortia and coordinators for providing these data and the clinical information used here. This study was partially funded by patient foundations in Catalonia (DACMA, GINKGO, "Viladecans Contra el Cáncer", and "Toca-te-les"), the "Instituto de Salud Carlos III" through the grants PI16/00563, PI18/01029, PI19/00553, PI21/01306, and CIBERONC and CIBERES (cofunded by the European Regional Development Fund. ERDF, a way to build Europe), the Generalitat de Catalunya (SGR 2017-449 and 2017-1282; and PERIS PFI-Salut SLT017-20-000076, Suport SLT017/20/000072, MedPerCan, and URDCat), La Marató de TV3 2019 (project 240), the CERCA Program to IDIBELL, the Canadian Institutes of Health Research (169111), the Canadian Cancer Society (BC-RG-16), the Michael Cuccione Foundation for Childhood Cancer Research, the BC Children's Hospital Research Institute (to C.A.M.), the Murray Family (to L.M.), the University of British Columbia 4-year fellowship program (to Z.H. and A.L.), and the National Institute of Health R01CA220578 grant (to R.L.).

## Author contributions

C.A.M. and M.A.P. conceived, designed, and supervised the study. F.M., G.R.G., C.H., N.G., and A.B. performed the mouse studies. F.M., G.R.G., C.H., N.G., A.G-U., and A.I.E. performed mouse genotyping. Z.H. performed the ex vivo cell studies and contralateral tissue analyses. L.M. performed the mechanistic studies of genome instability and the contralateral tissue analyses. G.R.G. and O.M. undertook the mouse fibroblast studies. F.M., Z.H., and L.M. performed the gene and protein expression analyses in mammary tissue and tumors. E.B., L.P., and R.E. performed the bioinformatic analyses. A.G. performed the RNA-seq data analysis of tumors. Z.H. and A.L. performed the RNA-seq data analyses of contralateral tissue. T.S. and A.P. carried out the histopathological characterization of tumors. P.F.L. performed mass spectrometry analysis. Z.H., L.M., K.C., S.T., and C.J.E. performed luminal progenitor studies. R.L., J.B., C.M., R.H., R.K., and C.L. contributed to data analysis and interpretation. F.M., Z.H., and L.M. helped write the first draft of the manuscript, and C.A.M. and M.A.P. wrote the final manuscript.

## Competing interests

M.A.P. received an unrestricted research grant from Roche Pharma to develop the ProCURE ICO research program. The other authors declare no competing interests.
