## [Peer Review File · Nature Communications]

Modification of BRCA1-associated breast cancer risk by HMMR overexpressionReviewers' Comments:

Reviewer #1:

Remarks to the Author:

Mateo, Pujana et al. describe bioinformatic and experimental studies that establish HMMR overexpression as a modifier of BRCA1 mutant breast cancer. This follows original work from Pujana in 2007 that identified HMMR overexpression as a factor that promotes breast cancer (Nature Genetics 2007). The current study from Mateo et al. examine this in a two part system, that demonstrates HMMR overexpression in vivo accelerates breast cancer when combined with BRCA1 mutation in the mouse mammary gland, and also increases hallmarks of cGAS-STING induced inflammatory signaling, including EMT and tissue infiltration with tumor associated macrophages.

This is a well constructed story that should be of interest to the breast cancer community. I have a few recommendations below that should be addressed prior to publication.

1. Fig 3a- Does HMMR overexpression alone increase mammary tumors? Would it affect latency in other models?
2. The authors stress the importance of cGAS-STING induced noncanonical NF- κ B activation to the pathogenesis of tumors arising in their crosses. However, cGAS-STING deficiency was not examined in the GEMM models, making this an associated finding rather than a causative one. This should be duly noted in the text.
3. Missegregation causing cGAS-STING was first shown by several groups (PMIDs:28738408; 28759889; 28633018) These papers should cite along with that of Bakhoun and Cantley

Reviewer #4:

Remarks to the Author:

Mateo and colleagues describe a large study that aims to understand the mechanisms that associate between BRCA and HMMR in the context of breast cancer risk. The manuscript includes multiple layers of analysis, from computational GWAS analyses to mechanistic elucidation of the function of HMMR. Overall, the manuscript would benefit from better structuring and separation to sections, as it is missing a proper background and would be helpful to summarize subsections to ease the reading. I focus my review on the MS analysis of HMMR interactions. In the current version of the manuscript it is impossible to assess the experiment and the analyses. The authors have to state what are the systems that were actually pulled down and detail the data analysis procedures. It is not clear whether HMMR was overexpressed for the pulldown, and compared to non-overexpressing cells, or whether pulldown was performed on the same cells with two different antibodies (HMMR and IgG). In fact, the writing of the results part and the method part suggest two different things. In case overexpressing cells were compared to controls, I suspect that some of their binders may have been found due to cellular changes that occurred due to the overexpression, and are not necessarily HMMR-binders. Furthermore pulldown of the actomyosin machinery is very common since these are large complexes with many binders. Once one member of this machinery is pulled down, often the whole system is pulled down together with it. One has to be very cautious in interpreting this, and aim to understand (and validate) the direct binders.

There is also missing information in the method section. What were DTT and IAA dissolved in? IAA should be corrected to iodoacetamide, and not indole-3-acetic acid.

While the pull-down itself is described in the methods, I couldn't find any information about the data analysis. What software was used? FDR? modifications? database? what statistical parameters were used to define the binders?

All of these are basic requirements necessary to provide a proper assessment of these experiments.

Reviewer #5:

Remarks to the Author:

Mateo et al present a comprehensive and rigorously conducted study on the role of HMMR/RHAMM in the progression of BRCA1 deficiency associated triple negative breast cancer. This study represents an in-depth followup into the mechanisms by which HMMR overexpression may promote tumor development and/or progression, which was initially reported by some of these authors in 2007. In this study, a novel transgenic mouse model with Cre-inducible expression of human HMMR in combination with Brca1 and p53 deficiency reveals causative roles for HMMR in tumor development and genome instability-associated phenotypes. Inducible overexpression in a cell line model demonstrates a role in micronuclei generation and cGAS/STING pathway activation. Proteomic analyses suggest a possibly direct role in mitotic spindle function. The authors utilize RNA-seq analyses of tumors and pre-malignant mammary glands, and also elegantly incorporate published single cell RNAseq datasets in an effort to reconstruct a series of observations into a temporal framework of tumorigenic transformation.

While the data included in this paper is generally compelling, I believe its presentation and interpretation can be improved. The activation of cGAS/STING signaling by HMMR may not have a causative role in tumorigenesis, and may very well be a byproduct of enhanced genome instability, which may be the driving force for transformation in a model that requires p53 loss-of-heterozygosity. The mechanism underlying HMMR roles in genome integrity is not explored in much detail. While cGAS/STING/NF-kb signaling can exert pro-tumorigenic effects, there are also well described tumor suppressive roles for these pathways that have not been discussed or considered. In general, I believe the condensed letter format does an injustice to the volume of data in this paper, and makes it hard for the reader to contextualize the advances presented here in relation to other information that is known about HMMR, and also lacks sufficient discussion regarding the remaining unanswered questions regarding its role in tumor progression.

Specific critiques:

- 1) Consider making the title and conclusion sentence in the abstract more specific to the role of HMMR in breast cancer progression. The current versions are vague and not reflective of the article content.
- 2) The analyses presented in Figure 1 are not very compelling and distract from the high quality of data/results presented in the remainder of the paper. These authors have already shown HMMR to be associated with BRCA1 mutant breast cancer. The correlation between HMMR and the BRCA1-associated gene sets is difficult to interpret and does not guide any of the future studies performed in this paper. My preference would be to remove these analyses from this otherwise excellent paper.
- 3) The transgenic mouse model utilizes human HMMR expression. It would be helpful to compare levels of the human transgene expression to endogenous mouse HMMR expression, if possible.
- 4) The observations reported in Fig 6 are due to Brca1 haploinsufficiency, and not necessarily due to HMMR expression. This should be clearly stated in the discussion of the results. I believe their data may argue that there is a regulatory relationship between Brca1 haploinsufficiency and HMMR expression levels that may promote genome instability. If this is true, then it would be interesting to examine phenotypes associated with HMMR knockdown in a cell line model with inducible Brca1 deficiency.
- 5) I would advise the authors to revise their conclusion/final paragraph. They report a series of observations that have not been mechanistically connected, but are presented as a sequence of events that "provide a comprehensive explanation for immune cell changes and a luminal progenitor to basal/mesenchymal transition before tumorigenesis". This statement is premature and likely inaccurate. The authors have uncovered a link between HMMR overexpression and tumor progression, a link between HMMR expression and cGAS activation, and so on with a few other observations. However, these cannot be linked together without further perturbations of cGas or other putative downstream mediators. Several unanswered questions regarding the mechanistic effects of HMMR expression remain to be addressed, and should be highlighted in a revised discussion section that states some of the limitations of the present study.

Minor Point:

1) There are typos in the call-outs to Figure 4e and 4f on page 11.

Reviewer #6:

Remarks to the Author:

It is important to identify genetic modifiers for BRCA1 mutation carriers and understand the biological process that underpin BRCA1-associated breast cancer. The study by Mateo et al. modeled Brca1-mutant mammary tumorigenesis and demonstrated that overexpression of HMMR in mice increases the penetrance of BRCA1 mutation (Figure 3a). The study showed that HMMR increases BRCA1-associated breast cancer by hindering ARPC2 localization in the mitotic cell cortex, inducing micronucleation, and activating non-canonical NF-KB signaling and EMT.

Major comments:

The hypotheses being tested in the gene-set analysis and abstract mining are unclear. Seemly these analyses are not directly relevant to the main study focus, the gene HMMR. The co-expression of HMMR and gene sets in TCGA (Figure 1f), more positives or more negatives, does not mean the HMMR gene can modify BRCA1-associated breast cancer risk.

In the genome-wide association study of CIMBA in BRCA1 mutation carriers and triple negative breast cancer, the C allele of rs299290 in the gene HMMR was only marginally associated with increased breast cancer risk (log relative risk = 0.0442, $p = 0.001$). Although it is not genome-wide significant, this should be reported in this paper.

Mateo F, He Z, Mei L *et al.*, "Modification of *BRCA1*-associated breast cancer risk by HMMR overexpression" (NCOMMS-21-19685-T, original: "Sequential molecular and cellular alterations increase risk of *BRCA1*-associated triple-negative breast cancer").

Reviewer #1 (Remarks to the Author)

Mateo, Pujana et al. describe bioinformatic and experimental studies that establish HMMR overexpression as a modifier of BRCA1 mutant breast cancer. This follows original work from Pujana in 2007 that identified HMMR overexpression as a factor that promotes breast cancer (Nature Genetics 2007). The current study from Mateo et al. examine this in a two-part system, that demonstrates HMMR overexpression in vivo accelerates breast cancer when combined with BRCA1 mutation in the mouse mammary gland, and also increases hallmarks of cGAS-STING induced inflammatory signaling, including EMT and tissue infiltration with tumor associated macrophages.

This is a well-constructed story that should be of interest to the breast cancer community. I have a few recommendations below that should be addressed prior to publication.

We are grateful for the positive comment about our study. We have tried to address the questions raised as described below.

1. Fig 3a- Does HMMR overexpression alone increase mammary tumors? Would it affect latency in other models?

We acknowledge that this point was insufficiently described in our original submission, although it was mentioned in the Results prior to **Fig 3a**. Overexpression of HMMR alone did not cause tumors nor atypical epithelial cell structures in the mammary glands of the given mouse background (**Supplemental Fig. S5**). Greater perturbation of HMMR expression/function alone might be required to cause carcinogenesis and/or HMMR only acts as a BRCA1-modifier in a p53-deficient setting. The reviewer's question is also relevant given that *HMMR* rs299290 was originally shown to be associated with risk of breast cancer in Ashkenazi Jewish individuals (PMID:17922014), including non-carriers of *BRCA1* pathological variants, but this association was not observed in non-Ashkenazi populations (PMID:19064580). This discrepancy could indicate the existence of additional genetic differences in the *HMMR* locus among Ashkenazi Jewish that further influence breast cancer risk: of note, the modifier effect of rs299290 was not different between Ashkenazi and non-Ashkenazi carriers of *BRCA1* pathological variants (PMID:22110403 and PMID:25830658). The Results (page 7, lines 163-167) and Discussion (pages 18-19, lines 426-430) sections have been edited to clarify these points.

2. The authors stress the importance of cGAS-STING induced noncanonical NF-kB activation to the pathogenesis of tumors arising in their crosses. However, cGAS-STING deficiency was not examined in the GEMM models, making this an associated finding rather than a causative one. This should be duly noted in the text.

The reviewer is correct that alteration of cGAS-STING signaling was not directly assessed *in vivo* and, therefore, that the original results and model-stages were correlative. In order to address this point, we performed additional experiments in which cGAS deficiency was instigated using a specific small-molecule inhibitor: RU.521 (PMID:28963528). Briefly, MECs were isolated from *Blg-Cre;Brca1^{ff};Trp53^{+/-}* mice or *Blg-Cre;HMMR^{Tg/Tg};Brca1^{ff};Trp53^{+/-}* mice, transduced with EGFP-Cre lentivirus, and sorted based on the expression of EGFP to ensure Cre-mediated loss of BRCA1 and overexpression of HMMR. The MECs were then treated with vehicle or one of two doses RU.521 (0.7 μ M or 1.4 μ M) for 48 hours, mRNA extracted, and expression of proinflammatory genes determined. The results of these assays showed inhibitor-mediated downregulation of 4/6 genes previously demonstrated to be overexpressed in *Blg-Cre;HMMR^{Tg/Tg};Brca1^{ff};Trp53^{+/-}* MECs: the new data is presented in **Supplemental Figure S27b** and the corresponding Results section has been edited (page 16, lines 373-378).

3. Missegregation causing cGAS-STING was first shown by several groups (PMIDs:28738408; 28759889; 28633018) These papers should cite along with that of Bakhoun and Cantley

We apologize for these omissions; the references of these seminal studies have been included in page 11 (lines 260-262).

Reviewer #4 (Remarks to the Author)

Mateo and colleagues describe a large study that aims to understand the mechanisms that associate between BRCA and HMMR in the context of breast cancer risk. The manuscript includes multiple layers of analysis, from computational GWAS analyses to mechanistic elucidation of the function of HMMR. Overall, the manuscript would benefit from better structuring and separation to sections, as it is missing a proper background and would be helpful to summarize subsections to ease the reading.

We thank the reviewer for the suggestions. The manuscript was originally composed in a report format; in this resubmission, the main sections are correctly presented and structured to ease reading and comprehension.

I focus my review on the MS analysis of HMMR interactions. In the current version of the manuscript it is impossible to assess the experiment and the analyses. The authors have to state what are the systems that were actually pulled down and detail the data analysis procedures. It is not clear whether HMMR was overexpressed for the pulldown, and compared to non-overexpressing cells, or whether pulldown was performed on the same cells with two different antibodies (HMMR and IgG). In fact, the writing of the results part and the method part suggest two different things. In case overexpressing cells were compared to controls, I suspect that some of their binders may have been found due to cellular changes that occurred due to the overexpression, and are not necessarily HMMR-binders.

The reviewer is right to raise these issues and we regret any difficulty that arose in interpreting the results. In the original manuscript, we performed IP-MS on lysates obtained from synchronized HeLa cells to gain insights into a general mitotic mechanism for HMMR. The corresponding Methods section (pages 28-29) has been edited.

Furthermore, pulldown of the actomyosin machinery is very common since these are large complexes with many binders. Once one member of this machinery is pulled down, often the whole system is pulled down together with it. One has to be very cautious in interpreting this, and aim to understand (and validate) the direct binders.

This is also a valuable concern and we thank the reviewer for raising it, giving us the opportunity to clarify the issue. We should have referenced prior independent work that identified HMMR (aka RHAMM or IHABP) as a binder to microtubule and actin filaments ([PMID:10547355](https://pubmed.ncbi.nlm.nih.gov/10547355/); now stated on page 14, line 316). We also noted that ARPC2 and HMMR share four interactors, as compiled in the BioGRID database (<https://thebiogrid.org/>: AURKA, LRRK2, NDC80, and PRKACA). Throughout cell division, HMMR localizes to the mitotic spindle and spindle poles ([PMID:12808028](https://pubmed.ncbi.nlm.nih.gov/12808028/)), but the ARP2/3 complex only localizes to spindle poles during anaphase ([PMID:31015335](https://pubmed.ncbi.nlm.nih.gov/31015335/)). However, endogenous co-immunoprecipitation assays of HMMR in HeLa cells did not show an association with the major constituent of the ARP2/3 complex, ACTR3 (new **Supplementary Fig. S20**), which could indicate that the proposed interaction is indirect and/or restricted to mitotic spindle poles at specific phase(s). Analysis of ARPC2 localization during mitosis confirmed a restricted recruitment to spindle poles during

metaphase-anaphase, with HMMR overexpression triggering earlier recruitment, in prophase-prometaphase (original **Supplementary Fig. S18**, now **S19**, and new **S21**). Thus, abnormal ARPC2 localization to the spindle poles may be due to direct binding to overexpressed HMMR and/or to an indirect unknown mechanism.

To further assess this point, we examined the consequences of inhibiting mitotic kinases whose activity is known to be promoted by HMMR expression: AURKA (PMID:24875404 and PMID:25532896), CSNK1A1 (PMID:31338967), and PLK1 (PMID:28994651). Inhibition of AURKA with 1 nM MLN8237 was able to normalize both mitotic blebbing and daughter cell size in HeLa cells overexpressing HMMR, but these phenotypic normalizations were not observed by inhibition of CSNK1A1 or PLK1 (new **Supplementary Fig. S22**). Then, evaluation of pThr488-AURKA signal intensity at mitotic pericentrin (PCTN)-positive centrosomes indicated overactivation of AURKA in HeLa cells that overexpress HMMR (new **Fig. 5k**). In addition, exposure to 1 nM MLN8237 reduced pThr488-AURKA intensity in HeLa mitotic cells (new **Fig. 5l**), and reduced spindle pole localization of ARPC2, with normalized cortical localization (**Fig. 5m**). These observations appeared to be consistent with frequent overexpression of AURKA and HMMR in *BRCA1*-mutant breast cancer cell lines and tumors (PMID:17922014 and PMID:22110403), and suggest that relative earlier localization of ARPC2 to mitotic spindle poles is mediated by overactivation of AURKA-HMMR. We have added these data and interpretation to a new Results section (pages 14-15) and the proposed model has been edited to include the additional player (**Fig. 7c**).

There is also missing information in the method section. What were DTT and IAA dissolved in? IAA should be corrected to iodoacetamide, and not indole-3-acetic acid. While the pull-down itself is described in the methods, I couldn't find any information about the data analysis. What software was used? FDR? modifications? database? what statistical parameters were used to define the binders? All of these are basic requirements necessary to provide a proper assessment of these experiments.

We thank the reviewer for highlighting these errors and omissions in the Methods section, and we sincerely regret any difficulty this caused in interpreting the results. The acquired spectra were searched against a *Homo sapiens* protein database using Proteome Discoverer employing 1% FDR cut-off at peptide and protein level. The following modifications were considered: static modifications, +57.021 Da on C; variable modifications, +15.995 Da on M, +28.031 or +34.063 on K; and peptide N terminus, +42.011 Da. Differential abundance in anti-HMMR versus rabbit IgG immunoprecipitates was determined using a Student's *t*-test and proteins commonly identified in affinity enrichment experiments were flagged based on information derived from the CRAPome database. We have edited and added these important details to the corresponding Methods section (pages 28-29).

Reviewer #5 (Remarks to the Author)

Mateo et al present a comprehensive and rigorously conducted study on the role of HMMR/RHAMM in the progression of BRCA1 deficiency associated triple negative breast cancer. This study represents an in-depth followup into the mechanisms by which HMMR overexpression may promote tumor development and/or progression, which was initially reported by some of these authors in 2007. In this study, a novel transgenic mouse model with Cre-inducible expression of human HMMR in combination with Brca1 and p53 deficiency reveals causative roles for HMMR in tumor development and genome instability-associated phenotypes. Inducible overexpression in a cell line model demonstrates a role in micronuclei generation and cGAS/STING pathway activation. Proteomic analyses suggest a possibly direct role in mitotic spindle function. The authors utilize RNA-seq analyses of tumors and pre-malignant mammary glands, and also elegantly incorporate published single cell RNAseq datasets in an effort to reconstruct a series of observations into a temporal framework of tumorigenic transformation.

While the data included in this paper is generally compelling, I believe its presentation and interpretation can be improved.

We thank the reviewer for commenting on the quality and interest of our study. We also acknowledge the limitations in the presentation, interpretation, and discussion of the results in our initial submission. These issues have been addressed through subsequent improvements made in response to the reviewer's comments and recommendations.

The activation of cGAS/STING signaling by HMMR may not have a causative role in tumorigenesis, and may very well be a byproduct of enhanced genome instability, which may be the driving force for transformation in a model that requires p53 loss-of-heterozygosity. The mechanism underlying HMMR roles in genome integrity is not explored in much detail. While cGAS/STING/NF-kb signaling can exert pro-tumorigenic effects, there are also well described tumor suppressive roles for these pathways that have not been discussed or considered. In general, I believe the condensed letter format does an injustice to the volume of data in this paper, and makes it hard for the reader to contextualize the advances presented here in relation to other information that is known about HMMR, and also lacks sufficient discussion regarding the remaining unanswered questions regarding its role in tumor progression.

The reviewer is correct that there were original caveats regarding the mechanistic link between HMMR overexpression and cGAS/STING signaling, and the proposed tumorigenic model. The revised manuscript has been extensively edited to address these issues and to include formal sections to ease comprehension.

Specific critiques:

1) Consider making the title and conclusion sentence in the abstract more specific to the role of HMMR in breast cancer progression. The current versions are vague and not reflective of the article content.

We acknowledge this concern and have edited the manuscript title and abstract to highlight the study of HMMR as a BRCA1-modifier.

2) The analyses presented in Figure 1 are not very compelling and distract from the high quality of data/results presented in the remainder of the paper. These authors have already shown HMMR to be associated with BRCA1 mutant breast cancer. The correlation between HMMR and the BRCA1-associated gene sets is difficult to interpret and does not guide any of the future studies performed in this paper. My preference would be to remove these analyses from this otherwise excellent paper.

We recognize the reviewer's concern about the introductory results to the study of HMMR as BRCA1-modifier but, respectfully, we would like to keep this section in a shortened and improved form, for the following reasons. Despite identifying many loci as modifiers of *BRCA1*-associated breast cancer risk, the biological process and signaling pathways, if any, in which they may converge is not fully understood. We believe this question is relevant to being able to eventually define the "core" of modifying processes/pathways and their interplay, which could more comprehensively clarify *BRCA1*-associated cancer biology. This question may also be considered appropriate in our study that evaluates the impact of one potential modifier (i.e., HMMR) across different biological levels.

The analysis of GWAS results indicates that there is diversity in the biological basis of risk, but notably a substantial portion of risk foundation appears to converge on perturbation of BRCA1-associated functions and interactors, including HMMR. It is of note that the predictions also included immune system processes/pathways, which are also noted in subsequent results. These observations are also consistent with the idea (also supported by the evidence obtained from single cell RNA-seq data in the original *Brcal^{fl/fl};Trp53^{+/-}* mouse model, **Supplementary Fig. S26**; and human bulk tumor profiles, **Fig. 7a**) that HMMR overexpression further perturbs processes that are altered by loss of BRCA1. We have considerably simplified **Fig. 1** and the corresponding analyses, and have tried to explain the reasoning and implications in the Results (pages 4-5) and Discussion (page 18, lines 407-412 and 422-424, including study limitations) sections.

3) The transgenic mouse model utilizes human HMMR expression. It would be helpful to compare levels of the human transgene expression to endogenous mouse HMMR expression, if possible.

This is also a good suggestion. TaqMan-based quantifications showed an approximated 1.3- and 2.4-fold overexpression of human *HMMR* relative to endogenous mouse *Hmmr* in developed tumors with the *HMMR^{+Tg}* and *HMMR^{Tg/Tg}* genotype, respectively, and about 1.9- and 2.1-fold overexpression in the analogous settings of mammary glands prior to tumor detection (new **Supplementary Fig. S8**, page 8).

4) The observations reported in Fig 6 are due to Brca1 haploinsufficiency, and not necessarily due to HMMR expression. This should be clearly stated in the discussion of the results. I believe their data may argue that there is a regulatory relationship

between *Bracl1* haploinsufficiency and HMMR expression levels that may promote genome instability. If this is true, then it would be interesting to examine phenotypes associated with HMMR knockdown in a cell line model with inducible *Bracl1* deficiency.

This is an accurate observation and links to the reasoning above regarding an “altered biological basis” that is further perturbed by HMMR overexpression. Indeed, *HMMR/HMMR* is frequently found to be overexpressed in *BRCA1*-mutant breast cancer cell lines and tumors (PMID:17922014 and PMID:22110403). We also described that concurrent depletion of *BRCA1* and HMMR impairs MCF10A acini formation in Matrigel 3D-culture (PMID:22110403). In additional experiments to address this comment, and using a MCF12A cell line with inducible Tet-On shRNA-*BRCA1*, we show that individual depletion of HMMR or *BRCA1* delays metaphase kinetics, but that concurrent depletion of these proteins does not normalize this mitotic alteration: see **Figure R1** for revision, below. The above interpretations have been included in the corresponding Results section (page 17, lines 398-401).

Fig. R1. Concurrent depletion of *BRCA1* and HMMR does not rescue the mitosis progression defect observed with each single *BRCA1*/HMMR perturbation. **a**, Representative images of immunofluorescence detection of HMMR (EGFP-tagged) and TUBB in Tet-On (-dox and +dox) shRNA-*BRCA1* (shBRCA1) MCF12A mitotic cells exposed to water or transduced with an siRNA against HMMR expression (siHMMR). **b**, Western blot analysis of HMMR depletion with siHMMR. **c**, Quantification of cell cycle phases in above MCF12A settings across time (0-60 minutes).

5) I would advise the authors to revise their conclusion/final paragraph. They report a series of observations that have not been mechanistically connected, but are presented as a sequence of events that “provide a comprehensive explanation for immune cell changes and a luminal progenitor to basal/mesenchymal transition before tumorigenesis”. This statement is premature and likely inaccurate. The authors have uncovered a link between HMMR overexpression and tumor progression, a link between HMMR expression and cGAS activation, and so on with a few other observations. However, these cannot be linked together without further perturbations of cGAS or other putative downstream mediators. Several unanswered questions regarding the mechanistic effects of HMMR expression remain to be addressed, and should be highlighted in a revised discussion section that states some of the limitations of the present study.

We agree with the reviewer. The Discussion section has been edited to recognize study limitations (pages 18-19, lines 422-439), while also comment on potential connections to recent observations made by others (page 19, lines 440-445).

Minor Point:

1) There are typos in the call-outs to Figure 4e and 4f on page 11.

We thank the reviewer for noting these mistakes, which have been corrected.

Reviewer #6 (Remarks to the Author)

It is important to identify genetic modifiers for BRCA1 mutation carriers and understand the biological process that underpin BRCA1-associated breast cancer. The study by Mateo et al. modeled *Brcal*-mutant mammary tumorigenesis and demonstrated that overexpression of HMMR in mice increases the penetrance of BRCA1 mutation (Figure 3a). The study showed that HMMR increases BRCA1-associated breast cancer by hindering ARPC2 localization in the mitotic cell cortex, inducing micronucleation, and activating non-canonical NF-KB signaling and EMT.

Major comments:

The hypotheses being tested in the gene-set analysis and abstract mining are unclear. Seemly these analyses are not directly relevant to the main study focus, the gene HMMR. The co-expression of HMMR and gene sets in TCGA (Figure 1f), more positives or more negatives, does not mean the HMMR gene can modify BRCA1-associated breast cancer risk.

We acknowledge the weaknesses in the presentation and interpretation of the data derived from gene set analyses of GWAS results, as also raised by reviewer #5. Despite the identification of many loci as modifiers of *BRCA1*-associated breast cancer risk, the potential convergence on defined biological processes and/or signaling pathways is not fully understood. We believe this is a relevant question within a study framework that evaluates the impact of one potential modifier (i.e., HMMR) across different biological levels. Our analysis of GWAS results indicates that there is diversity in the biological basis of risk, but also that a substantial portion of the risk foundation might converge on the perturbation of BRCA1-associated functions and interactors, including HMMR. The predictions also included relevant immune gene sets, as noted in subsequent Results sections. These observations are consistent with the idea (also supported by the evidence obtained from single cell RNA-seq data in the original *Brcal^{fl/fl};Trp53^{+/-}* mouse model, **Supplementary Fig. S26**; and human bulk tumor profiles, **Fig. 7a**) that HMMR overexpression further perturbs processes that are altered by the loss of BRCA1. However, we recognize the limitations about testing the modifier effect of different processes. We have considerably simplified **Fig. 1** and the corresponding analyses, and have tried to explain the reasoning and implications in the Results (pages 4-5) and Discussion (page 18, lines 407-412 and 422-424, including study limitations) sections.

In the genome-wide association study of CIMBA in BRCA1 mutation carriers and triple negative breast cancer, the C allele of rs299290 in the gene HMMR was only marginally associated with increased breast cancer risk (log relative risk = 0.0442, p = 0.001). Although it is not genome-wide significant, this should be reported in this paper.

This is also a valid point. The corresponding section has been edited to explain in more detail previous studies and results on this regard (page 5, lines 111-114).

Reviewers' Comments:

Reviewer #4:

Remarks to the Author:

The authors addressed all my comments. The manuscript, in its current form is much more structured and clear. The details of the MS experiments are also consistent and appropriate. There are still several missing points that should be added to the methods and the figure legend.

1. Missing information about the t-test. Is it a 1-sided test (seems like from the figure)? what is the FDR used as a cutoff?
2. Legend of Figure S17 should also indicate the differences in highlighted points (black vs. grey and different shapes and sizes)
3. The appearance of S100A(7-9) proteins in the IP should be addressed in text (results or discussion), as it could be directly related to other observations mentioned by the authors.

Reviewer #5:

Remarks to the Author:

The revised manuscript is significantly improved--both with additional data to strengthen the claims as well as more effective organization and description of results. All of the reviewer comments seem to be addressed adequately, and I support acceptance for Nature Communications. One minor point that needs to be addressed is pertaining to the hazard ratio and p values for OS and PFI in Figure 2b -- they are identical but I suspect this may be a typo.

Reviewer #6:

Remarks to the Author:

No additional comments.

Mateo F, He Z, Mei L *et al.*, "Modification of *BRCA1*-associated breast cancer risk by HMMR overexpression" (NCOMMS-21-19685A).

Reviewer #4 (Remarks to the Author)

The authors addressed all my comments. The manuscript, in its current form is much more structured and clear. The details of the MS experiments are also consistent and appropriate.

We are very grateful for the recognition of the improvements made in our resubmission, and we thank the reviewer for the initial suggestions and indications.

There are still several missing points that should be added to the methods and the figure legend.

- 1. Missing information about the t-test. Is it a 1-sided test (seems like from the figure)? what is the FDR used as a cutoff?**
- 2. Legend of Figure S17 should also indicate the differences in highlighted points (black vs. grey and different shapes and sizes)**

We regret these omissions. The statistical analysis and units represented in the plot shown in Supplementary Figure 17, and the symbol formats and colors, have been clarified in the corresponding legend. We have also provided all source data corresponding to this figure (including protein, peptide, and peptide spectrum matches) and have deposited the data in Proteome Exchange (ID PXD031752) and MassIVE (ID MSV000088870: doi:10.25345/C57659F06).

- 3. The appearance of S100A(7-9) proteins in the IP should be addressed in text (results or discussion), as it could be directly related to other observations mentioned by the authors.**

We thank the reviewer for observing these additional candidate interactors with HMMR. We haven't examined S100A7-9 in further detail, but we acknowledge their relevance in regulation of immune system homeostasis and inflammatory responses. S100A are calcium-binding proteins, and we also identified calmodulin-like 5 (CALML5) in HMMR co-IP results (Supplementary Figure 17), and HMMR was previously shown to bind calmodulin in a calcium-dependent manner ([PMID:10547355](https://pubmed.ncbi.nlm.nih.gov/10547355/)). These observations have been included in the Discussion section (pages 19-20).

Reviewer #5 (Remarks to the Author)

The revised manuscript is significantly improved--both with additional data to strengthen the claims as well as more effective organization and description of results. All of the reviewer comments seem to be addressed adequately, and I support acceptance for Nature Communications.

We are thankful for noticing the improvements made in our study, and also thank the reviewer for the initial advices.

One minor point that needs to be addressed is pertaining to the hazard ratio and p values for OS and PFI in Figure 2b -- they are identical but I suspect this may be a typo.

We thank the reviewer for noting this error. The corresponding HRs, 95% CIs and *p* values have been corrected.

Reviewer #6 (Remarks to the Author)

No additional comments.